# Emergence of a carbapenem-resistant atypical uropathogenic *Escherichia coli* clone as an increasing cause of urinary tract infection

Lachlan L. Walker[1,2], Minh-Duy Phan [1,2], Budi Permana[1,2], Zheng Jie Lian[1,2], Nguyen Thi Khanh Nhu [1,2], Thom Cuddihy[1,2], Kate M. Peters[1,2], Kay A. Ramsay [3], Chelsea Stewart[1,2], Niels Pfennigwerth[4], Timothy J. Kidd[5], Patrick N. A. Harris [3,5], David L. Paterson[6], Matthew J. Sweet [1,2], Brian M. Forde [1,2,3] ✉ & Mark A. Schembri [1,2,7] ✉

Carbapenem-resistant *Enterobacterales* pose a critical global health threat, exemplified by increasing resistance of uropathogenic *E. coli* (UPEC) that cause urinary tract infections (UTIs). Here, we investigate the publicly available EnteroBase dataset and identify a signal of increasing UTI caused by phylogroup A *E. coli* sequence type 167 (ST167). Phylogenetic analysis of ST167 based on whole genome sequence data reveal three major clades (A, B, C), with clade C further resolving into several subclades, notably subclade C2 that possessed high carriage rates of carbapenem and cephalosporin resistance genes. Hierarchical clustering of core genome multi-locus sequence typing reveals ~77% of subclade C2 strains contain <20 allelic differences in 2,513 core genes and harbour two distinct group 1 capsule types, KL124 and KL30, likely originating from *Klebsiella*. Subclade C2 was predominantly sequenced in North America, and we provide evidence for clonal expansion since 2016. Analysis of UPEC virulence factors reveals complete loss of the type 1 fimbriae genes in strains from clades B and C. Two subclade C2 isolates exhibit significantly reduced capacity to colonise the bladder compared to the reference UPEC strain CFT073 in a murine UTI model. Collectively, our data identifies ST167 as an atypical UPEC clone associated with UTI.

Urinary tract infections (UTIs) are one of the most common bacterial infections worldwide, with an estimated ~404 million cases per year and an annual economic burden of US$3.5 billion in the United States alone[1,2]. UTIs include infection of the bladder (cystitis), kidneys (pyelonephritis), and can lead to bacterial dissemination into the bloodstream, resulting in life-threatening urosepsis[2]. While antibiotics remain the standard mode of treatment, their widespread use has driven the emergence of resistant strains, particularly among

[1]Institute for Molecular Bioscience, The University of Queensland, Brisbane, QLD, Australia. [2]Australian Infectious Diseases Research Centre, The University of Queensland, Brisbane, QLD, Australia. [3]UQ Centre for Clinical Research, Faculty of Health, Medicine and Behavioural Sciences, The University of Queensland, Brisbane, Queensland, Australia. [4]German National Reference Centre for Multidrug-resistant Gram-negative Bacteria, Department of Medical Microbiology, Ruhr-University Bochum, Universitätsstraße 150, 44801 Bochum, Germany. [5]Pathology Queensland-Central Microbiology Laboratory, Queensland Health, Brisbane, QLD, Australia. [6]ADVANCE-ID, Saw Swee Hock School of Public Health, National University of Singapore, Singapore, Singapore. [7]School of Chemistry and Molecular Biosciences, The University of Queensland, Brisbane, QLD, Australia. ✉e-mail: b.forde@uq.edu.au; m.schembri@uq.edu.au

uropathogenic *E. coli* (UPEC), the causative agent of ~70% of UTIs worldwide[2,3]. Antibiotic resistance to first-line antibiotics such as trimethoprim, co-trimoxazole, ciprofloxacin, and nitrofurantoin, as well as third-generation cephalosporins and last-line carbapenems is of particular concern[4].

*E. coli* can be classified into eight broad phylogroups[5], with phylogroups B2 and D predominantly associated with extra-intestinal infections that include UTI[6]. *E. coli* strains can be further differentiated at higher resolution using multi-locus sequence typing (MLST)[7]. This scheme has identified key UPEC sequence types (STs; also referred to as clones), including ST131, ST1193, ST95, ST73 and ST69, which are responsible for a large proportion of UTIs and bloodstream infections globally[8–10]. These UPEC clones exhibit differences in their antibiotic resistance profiles, with ST131 and ST1193 exhibiting extensive resistance to multiple antibiotic classes[8]. The global emergence and dissemination of ST131 exemplifies the clinical impact of these clones[11]. ST131 possesses high levels of resistance to fluoroquinolones and third-generation cephalosporins and has become the most common cause of Gram-negative extra-intestinal infections since the turn of the 21st century[10,11].

In contrast to these UPEC lineages, ST10 and the ST10 clonal complex (ST10cc) belong to *E. coli* phylogroup A, a group typically associated with intestinal commensal colonisation[5]. The ST10cc includes clones such as ST10, ST167, ST617 and several other STs[12]. While *E. coli* strains from the ST10cc are not typically considered a major cause of extra-intestinal infections, there are reports of STs found in the ST10cc that exhibit high rates of carbapenem resistance and cause UTI and bloodstream infections[10,13,14].

In this study, we analysed the EnteroBase[15] dataset and identified an increasing number of ST167 genomes from UTI/urine samples between 2020-2024. Phylogenetic analysis of a subset of 800 ST167 genomes defined the clade structure of this high-risk clone and identified an emergent subclade (which we refer to as subclade C2) with very high carriage (89%) of genes encoding resistance to carbapenems and cephalosporins. In silico serotyping of ST167 genomic data revealed the acquisition of two group 1 capsule types within subclade C2, namely KL124^Group1 (KL124^G1) and KL30^Group1 (KL30^G1). We also found that ST167 strains carry significantly reduced numbers of UPEC virulence genes compared to ST131 and other UPEC strains, including an absence of genes encoding type 1 fimbriae in the majority of ST167 strains. Together, our analyses provide insight into the evolution of this emerging uropathogenic clone and identify distinct genomic signatures that support the classification of ST167 as an atypical UPEC clone associated with UTI.

## Results
### Identification of ST167 as a frequently sequenced clone associated with human UTI/urine

We screened EnteroBase[15] to detect signals of emerging *E. coli* lineages and found distinct patterns in the proportion of *E. coli* STs that cause UTIs. Data from 2024 (database accessed October 4th, 2024) revealed that ST167 represented the highest number of sequenced isolates from human UTI/urine (96 genomes), followed by ST410[16] (61 genomes) and ST131[11] (50 genomes) (Supplementary Fig. 1; Supplementary Table 1). The two most frequently sequenced STs from all isolate sources were ST10 and ST167[17], both of which belong to phylogroup A and the ST10cc[12]. Strains from *E. coli* phylogroup A are typically associated with intestinal commensal colonisation[5]. ST10 showed minimal association with UTI/urine (4/371 isolates). In contrast, almost half of ST167 isolates were from UTI/urine sources (96/197 isolates), higher than the pandemic UPEC ST131 clone (50/136 isolates) (Fig. 1a). Longitudinal analysis of UTI/urine isolates in EnteroBase from the five most deposited STs revealed a recent increase of ST167. ST167 increased ~17-fold in the EnteroBase dataset, from 1.2% of total UTI/urine isolates in 2020 to 21.2% in 2024 (Fig. 1a; Supplementary Table 2). The increase in

ST167 depositions to EnteroBase, coupled with the high number of sequenced isolates from UTI/urine sources, suggests this ST represents an emergent phylogroup A uropathogenic clone.

### Genomic phylogeny of ST167 reveals an emergent subclade with increased antibiotic resistance

To better understand the evolution and recent expansion of ST167, we analysed 2,066 ST167 genomes from EnteroBase[15]. Initial phylogenetic analysis with Mashtree[18] revealed five putative clades in the ST167 phylogeny (Supplementary Fig. 2). Subsequently, we constructed a higher resolution maximum likelihood phylogenetic tree using a randomly selected subset of 800 ST167 genomes, based on 109,894 recombination-free core-genome polymorphic sites (Fig. 1b). Bayesian analysis of population structure (fastbaps)[19] identified 17 distinct clusters at BAPS level 1 and level 2 (Supplementary Fig. 3). The three largest clusters were defined as clade A (70 genomes), B (314 genomes) and C (329 genomes), with the 14 remaining clusters ranging in sizes from 1–23 genomes (Supplementary Fig. 3; Supplementary Data 1). Clade C was further resolved into six subclades; subclade C1 and subclade C2 were the largest with 38 genomes and 262 genomes, respectively (Supplementary Fig. 3; Supplementary Data 1). Notably, 84% of subclade C2 strains were isolated after 2020 (Fig. 1c), while only 34% of clade B strains were isolated in the same period, suggesting the recent emergence of subclade C2 within the ST167 population. Further analysis of the isolation source revealed that subclade C2 had a significantly higher association with UTIs and urine samples (109/262 isolates [41.6%]) compared to clade A (10/70 isolates [14.3%]; $P = 6.8 \times 10^{-5}$), clade B (60/314 isolates [19.1%]; $P = 3.1 \times 10^{-8}$) and subclade C1 (7/38 isolates [18.4%]; $P = 4.16 \times 10^{-2}$; Fisher's exact test with Bonferroni-adjusted $p$-values) (Fig. 1d). This suggests that isolates from subclade C2 may exhibit increased pathogenic potential, providing valuable insights into the evolving epidemiology and potential clinical implications of the ST167 lineage.

The carriage of genes conferring resistance to antibiotics, including cephalosporins and last line carbapenems, is of clinical importance for treatment options. Among genes that conferred resistance to these classes of antibiotics, $bla_{CTX-M-15}$ (1186/2066) and $bla_{NDM-5}$ (1330/2066) were most common in ST167 genomes (Supplementary Data 2). Seventy four percent of genomes in clade B and 89% in subclade C2 possessed genes conferring resistance to both cephalosporins and carbapenems (Fig. 1e). To validate the in silico resistance profiles, we sourced strains from clade B ($n = 6$) and subclade C2 ($n = 5$) and conducted susceptibility testing for 24 antibiotics across 9 different classes. Most isolates from clade B and C2 were non-susceptible to cephalosporins and carbapenems, while there were differences in susceptibility for the isolates examined to amikacin, gentamicin and minocycline (Supplementary Data 3). All isolates examined were susceptible to nitrofurantoin (Supplementary Data 3).

### ST167 lack many UPEC-associated virulence factors, including type 1 fimbriae

To investigate the virulence potential of ST167, we conducted a comprehensive analysis of UPEC virulence genes across 800 genomes from our phylogenetic dataset. We curated a list of 74 non-redundant marker genes for established UPEC virulence factors, encompassing genes involved in adhesion, iron acquisition, immune evasion, motility and toxin production (Supplementary Data 4). We first calculated a virulence factor score for ST167 and phylogroup B2 and D STs strongly associated with UTI and demonstrated that ST167 has a reduced number of virulence factors compared to these UPEC clones (Supplementary Fig. 4). Comparative genomic analyses between ST167 and ST131 revealed that ST167 had significantly fewer UPEC virulence factors corresponding to chaperone-usher fimbriae, autotransporters, iron uptake, toxins and immune evasion mechanisms (Supplementary Fig. 5). The distribution of virulence factors within the phylogenetic

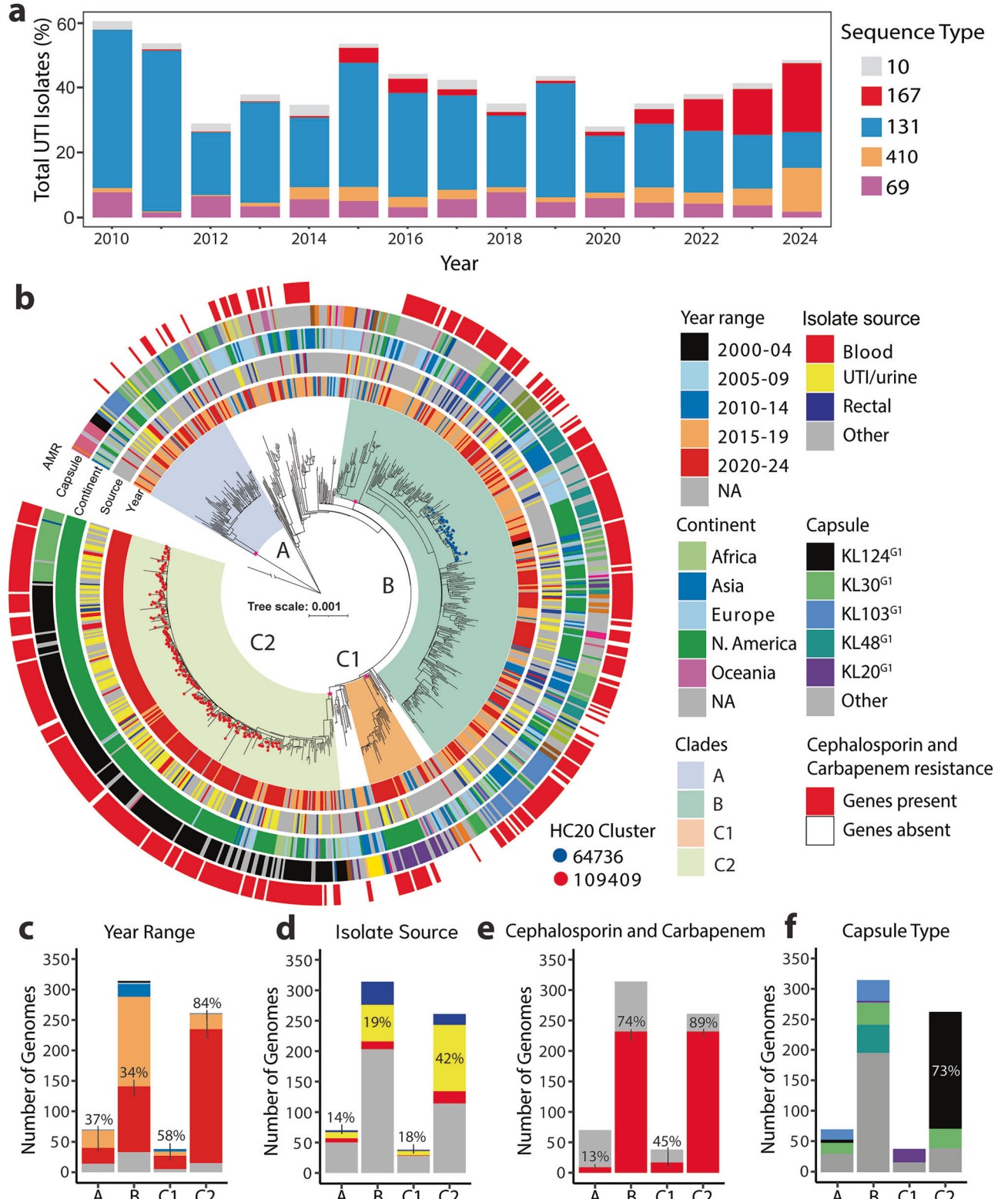

**Fig. 1 | Genomic analysis of ST167. a** Bar graph displaying the human UTI/urine isolate proportion (%) from 2010 to 2024 for the five most sequenced *E. coli* STs deposited to EnteroBase in 2024 (source data in Supplementary Table 2). **b** A Maximum likelihood tree of 800 ST167 genomes. The tree was constructed from 109,894 recombination-free core-genome polymorphic sites by IQ-TREE (GTR + G model and 1000 ultrafast bootstraps; with CP083869.1 used as reference and ED1a as outgroup). Tree scale is representative of nucleotide substitutions per site. ST167 clades A, B and subclades C1 and C2 are shaded and were determined using fastbaps. The HC20 clusters are coloured on tip points of the branches. The pink stars

on the branches of the major clade are indicative of >90% bootstrap support. The capsule type of the ST167 strains was determined using Kaptive 2.0. Antibiotic resistance genes carried by the ST167 strains was determined using AMRfinderplus, with red on the outermost ring indicating carriage of both carbapenem and cephalosporin resistance genes. **c**–**f** EnteroBase metadata was compiled to display bar charts for (**c**) collection year, (**d**) isolate source, 10/70 isolates were from UTI/urine sources for clade A, 60/314 for clade B, 7/38 for subclade C1 and 109/262 for C2, (**e**) carbapenem and cephalosporin resistance, and (**f**) capsule type.

structure of ST167 also revealed clade-specific patterns; for example, the presence of a specific *agn43* (antigen 43) allele in subclade C2 and absence of the yersiniabactin receptor gene *fyuA* in subclade C2 (Fig. 2a). Analysis of chaperone-usher fimbriae revealed the presence of genes encoding Ecp/Mat fimbriae (*ecpC*), Yad fimbriae (*yadV*) and Yfc fimbriae (*yfcV*) in the majority of strains, and Yeh fimbriae (*yehC*) in all ST167 except clade A strains. Strikingly, almost all ST167 strains except those from clade A lacked type 1 fimbriae (*fimD*). Further investigation into the absence of type 1 fimbriae genes revealed a complete replacement of the entire *fim* locus by the IS*1A* insertion element (Fig. 2b). Type 1 fimbriae are a critical UPEC virulence factor required for bladder colonisation[20,21]. Type 1 fimbriae-mediated binding via the tip-located FimH adhesin facilitates UPEC adhesion to α-D-mannosylated glycoproteins such as uroplakins on the bladder epithelium[22], facilitating invasion of bladder superficial epithelial cells[23] and the formation of intracellular bacterial communities (IBCs)[24]. To assess whether ST167 subclade C2 could colonise the bladder despite the lack of type 1 fimbriae, we employed a murine UTI model. ST167 subclade C2 strains MS25298 and MS25303 were recovered at variable loads in the urine, with bacteria detected in 11/19 and 5/19 mice, respectively. In contrast, the reference UPEC strain CFT073 belonging to ST73 was recovered in significantly higher bacterial loads in the urine (Fig. 2c). Bladder colonisation of MS25298 and MS25303 was also significantly reduced compared to CFT073; MS25298 and MS25303 were detected in 1/19 and 7/19 mice at ~10^4 CFU/g tissue, respectively, compared to CFT073 which effectively colonised the bladder of all mice at ~10^7 CFU/g tissue (Fig. 2d). Neither the ST167 strains nor CFT073 effectively colonised the kidneys (Fig. 2e), consistent with previous data from our lab using CFT073 for infection of C57BL/6 mice[25,26]. Taken together, our discovery that type 1 fimbriae are absent in most ST167 leads us to propose ST167 as an atypical UPEC clone associated with UTI.

### Core genome MLST reveals two closely related ST167 clusters

Evidence for the recent expansion of ST167 subclade C2 prompted a closer investigation into the genomic similarity of the ST167 genomes. To assess this, we extracted the hierarchical clusters of core gene multi-locus sequence typing (HierCC)[27] data at the HC20 level (reflecting genomes clustered with <20 allelic differences in 2513 core genes) from EnteroBase[15]. The ST167 population revealed two HC20 clusters that contained >100 genomes each (Fig. 1a). Clade B contained the HC20 cluster 64736, which was primarily isolated from companion animals (75/176 isolates). Subclade C2 contained HC20 cluster 109409 which consisted almost exclusively of isolates from humans (419/433; Fig. 3a, b). Strains in the two HC20 clusters possessed high rates of carriage of *bla*~NDM-5~ (carbapenem resistance), *bla*~CTX-M-15~ and *bla*~OXA-1~ (cephalosporin resistance) (Fig. 3c). In contrast, carriage of other antibiotic resistance genes differed between the HC20 clusters (Fig. 3c). HC20 cluster 64736 showed high proportions of *bla*~TEM-1~ (resistance to penicillins), *dfrA*17 (trimethoprim resistance), *aac*(3)-lld (gentamicin resistance), as well as *aadA*5 and *aadA*22 (streptomycin resistance). Cluster 109409 contained a higher proportion of *sul*2 (sulfonamide resistance), *aph*(3')-la (kanamycin resistance), *aph*(6)-ld and *aph*(3")-lb (streptomycin resistance) genes.

### Serotype diversity within ST167 is determined by changes in capsule and flagella antigens

In silico serotyping analysis revealed differences in the capsule (K)- and flagella (H)-antigen among the HC20 clusters, while the O89b O-antigen was shared by strains in both HC20 clusters and the majority of other ST167 strains (Supplementary Fig. 6a). The O89b O-antigen is associated with group 1 capsule in *E. coli*[28,29]. Given the high levels of similarity between the group 1 capsule of *E. coli* and *Klebsiella pneumoniae*[30], we employed Kaptive 2.0[31], a *K. pneumoniae* capsule typing tool. Capsule typing identified 30 group 1 capsule types in

ST167 (Fig. 1f; Supplementary Data 2). The group 1 capsule is located between *galF* and *ugd* on the chromosome; the dominant HC20 cluster 109409 primarily contained the KL124^G1 capsule type (Fig. 3d) and a smaller proportion of the KL30^G1 capsule type (Supplementary Fig. 6b). HC20 cluster 64736 contained the KL48^Group1 (KL48^G1) capsule type (Supplementary Fig. 6b) that has also been identified in carbapenem resistant ST167[32]. H-antigen typing revealed HC20 cluster 64736 contained the H9 H-antigen, while 109409 harboured the H10 H-antigen (Supplementary Fig. 6a). These data demonstrate that subclade C2 is represented by a HC20 cluster with two distinct serotypes, KL124^G1:O89b:H10 and KL30^G1:O89b:H10.

Commonly, serotype diversity is determined by recombination events at the capsule, O-antigen and flagella loci. Acquisition of new genomic features through recombination has previously been linked to the expansion of other UPEC clones[33,34]. To investigate if ST167 subclade C2 acquired new genomic features via recombination we assessed recombination events using Gubbins[35]. This analysis identified 11 recombination regions >10 kb associated with subclade C2. The largest recombination region was ~387 kb in length and contained the flagella and capsule genes (region E; Supplementary Fig. 7). Other recombination regions contained genes associated with phage, genomic islands and core genes, including regions B (150 kb) and C (86 kb) that were restricted to ST167 subclade C2 (Supplementary Data 5).

### Bayesian analysis of ST167 subclade C2 identifies a rapidly expanding cluster comprising two distinct capsule types

The geographical distribution of subclade C2 shows that 87.6% of the strains were isolated in North America, associated with the expansion of the HC20 cluster 109409 which composed ~77% of subclade C2 isolates (Fig. 4a). To assess the population dynamics of subclade C2 we extracted all subclade C2 genomes from EnteroBase (*n* = 539) and assessed the temporal signal with TempEST[36]. There was evidence for a modest clock-like evolution (correlation coefficient = 0.3252; $R^2$ = 0.1057). The identification of the clock-like structure of subclade C2 was constrained by limited metadata detailing collection date, with only year of isolation consistently available across isolates. Despite these constraints, the positive temporal signal allowed us to employ BEAST[37] to reconstruct the evolutionary timeline of subclade C2. We determined the best-fitting model to be the uncorrelated relaxed exponential clock with the Bayesian skyline population model based upon the mean tree likelihood (Supplementary Data 6). The time to most recent common ancestor (TMRCA) of the HC20 cluster was 2015.8 (95% highest posterior density (HPD): 2014.9–2016.5) (Fig. 4a). The expansion of the HC20 cluster 109409 correlated with the Bayesian skyline plot that showed a clonal expansion around 2016 (Fig. 4b). Taken together, these data suggest HC20 cluster 109409 is an expanding clone with two distinct serotypes, primarily sequenced in the United States and associated with UTI.

## Discussion

Here we report on ST167 as an emergent carbapenem-resistant atypical UPEC clone associated with UTI. By generating a comprehensive ST167 phylogeny, we identified the acquisition of group 1 capsule genes and the loss of type 1 fimbriae genes as key features of most ST167 strains.

This study had several limitations. The genomic data was sourced from EnteroBase, a publicly available database that primarily obtains its data from the NCBI Short Read Archive (SRA). Although EnteroBase offers valuable insights into genomic epidemiology, its use of publicly submitted sequencing data reflects the sequencing activities and priorities of contributing researchers rather than a structured surveillance system. Consequently, the data does not provide a comprehensive representation of all bacterial populations or geographic regions, which may affect the generalisability of the study's findings.

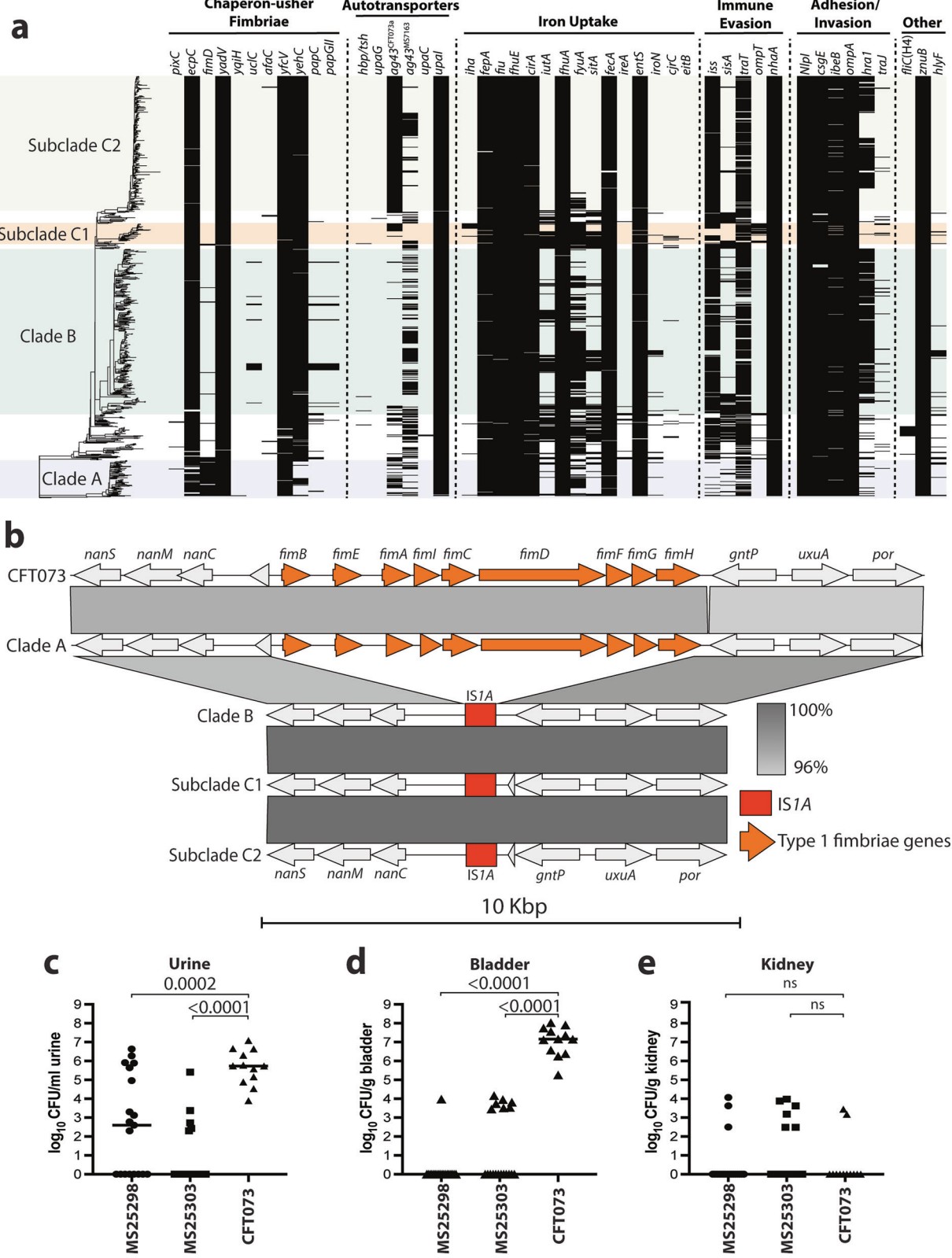

**Fig. 2 | Virulence analysis of ST167. a** Virulence factor heatmap of ST167. On the left is the phylogenetic tree of ST167 depicting the 800 genomes shown in Fig. 1a. The heatmap shows the presence and absence of the virulence factor marker genes; black indicates the virulence factor marker gene is present. The defined clades of the ST167 phylogeny are shaded. **b** EasyFig comparison of the type 1 fimbriae locus in CFT073 and representative complete genomes from clades A, B, C1 and C2. **c–e** Bacterial loads recovered from the (**c**) urine, (**d**) bladder and (**e**) kidneys of infected C57BL/6 mice expressed as median bacterial load per gram of tissue or ml of urine. Data are pooled from 2 independent experiments ($n = 19$ for strains MS25298 and MS25303; $n = 12$ for CFT073). Each symbol represents the bacterial load from an individual mouse at 24 h post infection. MS25298 and MS25303 are subclade C2 genomes from HC20 cluster 109409 and contain the KL124 capsule. Statistical significance was determined by an ordinary one-way ANOVA with the Dunnett correction applied; MS25298 and MS25303, respectively, were compared to CFT073.

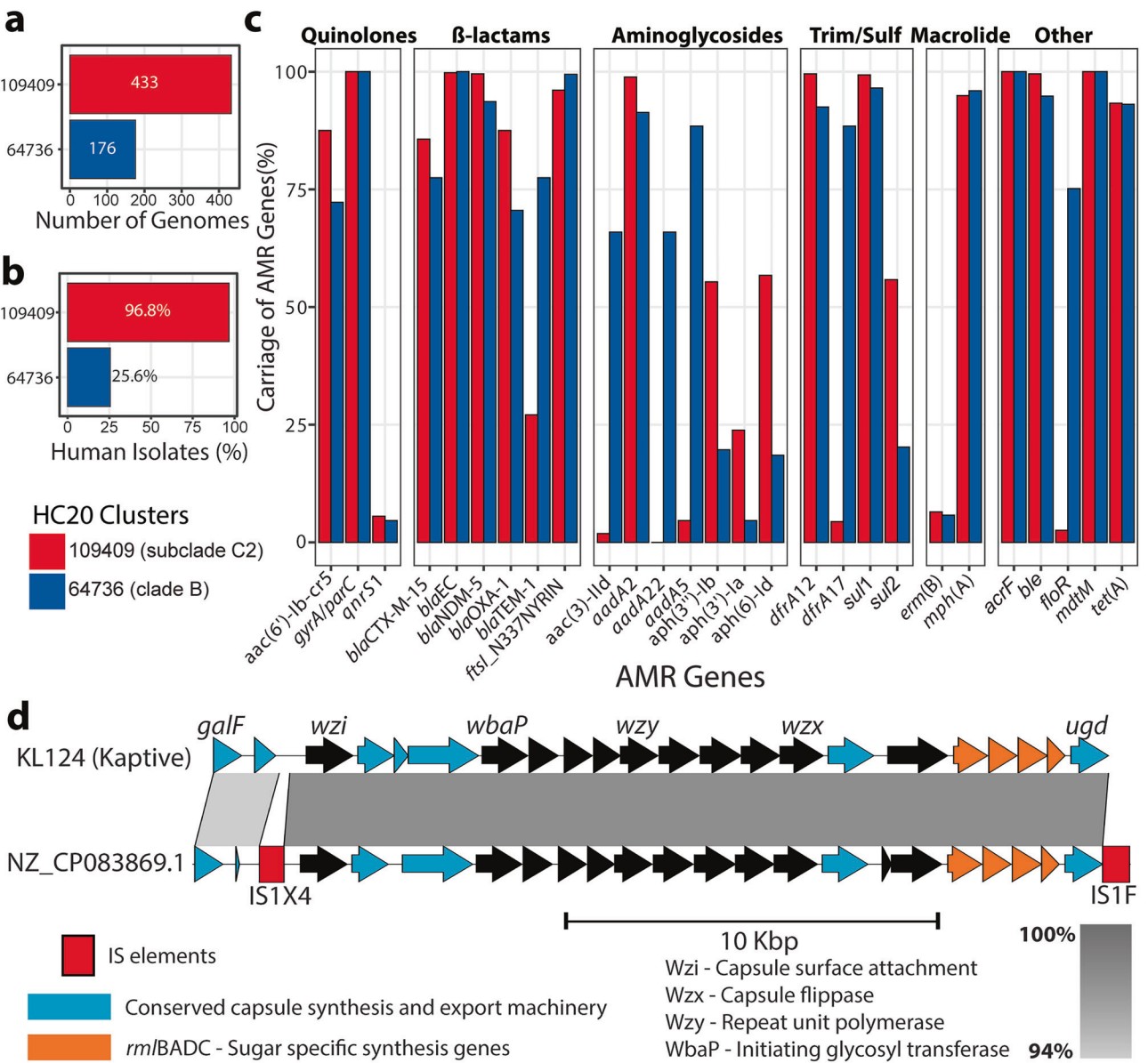

**Fig. 3 | Antibiotic resistance and capsule loci analysis of ST167 HC20 clusters. a** Bar graph depicting the number of genomes within each HC20 cluster. The colour is representative of HC20 cluster, red is 109409 and blue is 64736. **b** Bar graph showing the percentage of genomes isolated from human sources from each HC20 cluster. **c** Bar graph demonstrating the percentage of genomes from each HC20 cluster that carry the respective antibiotic resistance genes displayed on the x-axis (source data in Supplementary Data 2). The category of Trim/Sulf is representative of trimethoprim and sulfonamide antibiotic classes. The annotation of *gyrA/parC* denotes the chromosomal mutations of *gyrA*_D87N, *gyrA*_S83L, *parC*_S80I and *parE*_S458A. **d** Sequence comparison of KL124[G1] Kaptive 2.0 reference and NZ_CP083869.1 as a representative strain, visualised with Easyfig; the level of nucleotide sequence conservation is indicated by shading based on the legend. Capsule locus genes were annotated using Kaptive reference gene locus tags. IS elements were identified with ISfinder.

We note that the majority of ST167 North American UTI/urine isolates (88.3%) were collected as part of the Centers for Diseases Control and Prevention (CDC) National Healthcare Safety Network (NHSN). The NHSN is a large programme that conducts surveillance for healthcare events across diverse facilities in the United States, including the collection of data and biological samples associated with healthcare-associated infections (HAIs) and antimicrobial resistance[38]. Leveraging the NHSN extensive surveillance data, we inferred an increase in the proportion of genome sequenced UTI-associated *E. coli* ST167 in North America. However, the high number of genomes contributed to Enterobase through this programme means there are sampling biases in the data, for example the overrepresentation of human pathogens that: (i) are sequenced from high resource settings, (ii) cause HAIs, and

(iii) possess resistance to critical antibiotics such as carbapenems. Thus, while our findings identify ST167 subclade C2 as a frequently sequenced UPEC clone that causes UTI, we do not have sufficient data to compare infection incidence to other global clones such as ST131. Despite this bias, our analysis supports a distribution of ST167 beyond North America, with detection of subclade C2 isolates from geographic regions spanning Europe, Africa and Asia (Supplementary Fig. 8; Supplementary Data 2).

The geographical pattern we observed reflecting an increasing number of ST167 genomes with carbapenem resistance isolated in North America is also supported by the NCBI pathogen detection tool. Here, SNP cluster PDS000063179.375 (*n* = 528 isolates), which encompasses HC20 cluster 109409 of subclade C2, shows while 86.4%

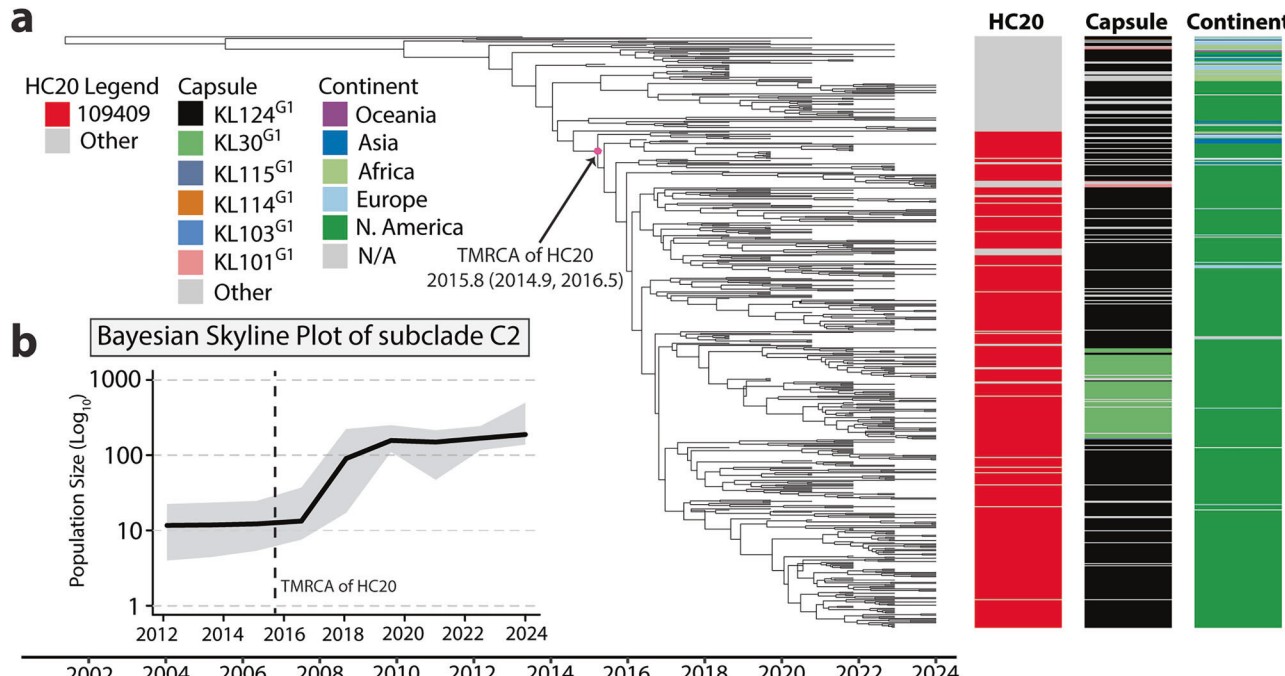

**Fig. 4 | Bayesian analysis of ST167 subclade C2. a** Time scaled tree of ST167 subclade C2 consisting of 539 genomes. The pink polygon depicts the node correlating to the TMRCA of HC20 cluster 109409. Metadata identifying HC20 cluster, capsule type and continent of isolation are displayed on the right as columns. **b** Bayesian skyline plot of subclade C2 genomes with the dotted line corresponding to the TMRCA of the HC20 cluster 109409. The black curve represents the effective population size, with the grey shading representing the 95% confidence interval.

of isolates originate from this region, isolates were also detected from 18 countries in other geographic regions (Supplementary Fig. 9). However, detailed assessment of infection incidence of subclade C2 in these other regions is limited by the availability and scope of comparable surveillance datasets. To further address potential database-driven biases, we conducted a meta-analysis of published studies from 2020 to 2025 that reported MLST data on carbapenem-resistant *E. coli* isolates, with a threshold of at least 50 isolates required for inclusion. The 33 studies included in this analysis varied widely in sample size and focus, with uneven geographical representation across regions, which limits the generalisability of the findings. Despite these constraints, the meta-analysis revealed that ST131 was the most frequently detected carbapenem-resistant clone, with ST167 detected at comparable levels to ST410, ST405 and ST38 (Supplementary Data 7).

In conclusion, our genomic investigation provides insights into the emergence of ST167 as a multidrug resistant UPEC clone associated with UTI, an observation that builds on other reports linking ST167 with increasing carbapenem resistance[13,39,40]. A striking finding from our genomic investigation of ST167 was the complete loss of genes encoding type 1 fimbriae in almost all ST167 strains from clade B, C1 and C2. Given the critical role of type 1 fimbriae for UPEC bladder colonisation in mouse models that replicate UTI[20,21], the inability to produce these fimbriae suggests ST167 disease pathogenesis is different to previously characterised phylogroup B2 and D UPEC strains and associated with alternative mechanisms of colonisation and/or virulence factors. The group 1 capsule identified in ST167, likely originating from *Klebsiella*, may be one such virulence factor. Our discovery also raises important considerations concerning ongoing UTI programmes aimed at the development of new anti-virulence drugs to block binding of the type 1 fimbriae FimH tip adhesin to the bladder uroepithelium[41], or vaccination with FimH[42], as such approaches would be ineffective at treating or preventing UTI caused by ST167. Together, these considerations form the basis of our criteria for the classification of ST167 as an atypical UPEC clone.

## Methods

### Ethics approval

Mouse infection experiments were approved by the University of Queensland Animal Ethics Committee (AEC approval number SCMB/259/19).

### EnteroBase data

The *Escherichia*/*Shigella* dataset from EnteroBase[15] was downloaded on the 4th of October, 2024. *Shigella* STs were identified and filtered out using a separate *Shigella*-specific dataset from EnteroBase[15]. To determine STs associated with UTIs, the *E. coli* dataset was first filtered for human isolates (search terms: 'Human|Homo sapiens') and subsequently for urinary sources (search terms: 'urine|UTI'). An ST with at least one isolate associated with human UTI identified using this method was considered a human UTI-associated ST. The human UTI-associated genomes were then selected from the entire dataset to identify the proportion of human UTI-associated STs. Data detailing the continent of origin for genomes and number of contributing sequencing projects per year is provided in Supplementary Data 8.

### Phylogenetic analysis

To construct a phylogeny for ST167, 2,104 draft genome assemblies were downloaded from EnteroBase[15]. Quality control was performed with QUAST[43] v5; genomes with >500 contigs and >20 undetermined bases per 100 Kb were removed, yielding a dataset of 2,066 genomes. A phylogenetic tree was constructed by Mashtree[18] v1.4.6 to understand the overall topology of ST167 (Supplementary Fig. 2).

To construct a maximum likelihood phylogenetic tree of ST167, 800 genomes were randomly selected in R. Prokka[44] v1.13 was then used for genome annotation. Core genome SNPs were called with snippy and snippy-core v4.6.0 (https://github.com/tseemann/snippy) generated a pseudo-genome alignment with the SNPs reinserted into the reference strain CP083869.1 genome (accession number CP083869.1); ED1a (accession number CU928162) served as an

outgroup for phylogenetic rooting. Regions of recombination were detected using Gubbins[35] v3.2.1 under default settings. Predicted recombinant regions were removed and the output was used to generate a phylogenetic tree with IQ-TREE[45] v2.1.2 using the general time-reversal GAMMA model (GTR + G) and 1000x ultrafast bootstraps[46], all other settings were default. Phylogenetic clade structure was determined using Bayesian analysis of population structure (fastbaps)[19] implemented in the R package fastbaps v1.0.8. Clade A was determined at the first level, clade B and C at the second level and subclades of clade C at the third level (Supplementary Fig. 3; Supplementary Data 1). The phylogenetic tree was visualised in iTOL[47].

### Accessory genome analysis

To determine the serotype of the ST167 genomes, we first performed capsule typing with Kaptive 2.0 software using the *Klebsiella* k locus primary reference database[31] (downloaded 27/09/24). Genomes with confidence of 'None' and 'Low' from the Kaptive output were considered untypeable. To confirm the similarity of group 1 capsule in *E. coli* and *Klebsiella*, we performed pairwise comparison between completely sequenced genomes from RefSeq and the Kaptive database reference loci using Easyfig[48] v2.2.2. The EcOH database (downloaded 10/08/2023) from the ABRicate software v1.0.1 (https://github.com/tseemann/abricate) was used to determine O-antigen and H-antigen type[49]. Analysis of antibiotic resistance gene carriage in ST167 genomes was performed with AMRfinderplus[50] v3.11.20. PHASTEST[51] was used to identify phage regions within the reference strain CP083869.1. Phandango[52] was used to visualise recombination regions. ISfinder[53] was used to identify IS elements.

### Virulence factor analysis

To identify UPEC-associated virulence factors we curated a list of established UPEC virulence factor marker genes from the literature (Supplementary Data 4). This virulence factor list contained 74 non-redundant marker genes. Comparative analysis with the ST131 was performed using a previously published ST131 dataset containing 3,993 genomes[54].

### Antibiotic susceptibility testing

Antibiotic susceptibility testing of ST167 isolates was performed using broth microdilution (Sensititre; Thermo Fisher Scientific) and results were interpreted according to European Committee on Antimicrobial Susceptibility Testing (EUCAST) clinical breakpoints criteria V13.0[55]. For antibiotics that do not have a defined breakpoint according to EUCAST guidelines (e.g. minocycline) we deferred to Clinical and Laboratory Standards Institute (CLSI) guidelines[56]. Nitrofurantoin susceptibility was determined using the breakpoint for uncomplicated UTI[55].

### Mouse infection experiments and antibiogram testing

All experiments were performed using the following housing conditions: (light:dark cycle 12:12 h, room temperature 21 ± 1 °C, humidity 50 ± 10%). The mouse UTI model was performed essentially as previously described[57,58]. Strains were cultured overnight in LB broth with shaking for inoculum preparation. Female C57BL/6 mice (8–10 weeks) were infected via transurethral inoculation at $5 \times 10^8$ CFU/ml. Mice were euthanised by cervical dislocation at 24 h post infection. Bacterial loads corresponding to each strain in the urine, bladder and kidney of infected mice were enumerated by plating onto LB agar. All experiments were conducted in duplicate using groups of 9-10 mice (MS25298 and MS25303) or 6 mice (CFT073). Strains MS25298 and MS25303 were tested in a total of 19 mice, respectively; CFT073 was tested in a total of 12 mice.

### Bayesian temporal analysis

The HC100 cluster 71026 from EnteroBase[27] was identified to encompass the entirety of subclade C2 genomes and was extracted. After

filtering with QUAST[43] there was a collection of 539 ST167 subclade C2 genomes with metadata for collection year. Preliminary estimation of the temporal signal was determined by TempEST[36] by performing a regression analysis of root-to-tip genetic distance extracted from a recombination-free maximum-likelihood phylogenetic tree. The IQtree[45] model selection function was used to identify the nucleotide substitution model for BEAST[37] and was determined to be a GTR + G model. The Gamma Site Model Category Count was set to four and the GTR substitution model rates determined from IQtree[45] were included (rate AC = 1.0715, AG = 2.9806, AT = 1.8727, CG = 0.1453, CT = 3.2344, and GT = 1.00). The initial clock rate was set to $5.3222 \times 10^{-4}$ as estimated by the root-to-tip regression analysis in TempEST[36]. To identify the optimal combination for molecular clock and population growth models, we evaluated a number of combinations with BEAST v2.6.7. These included the strict, log-normal and relaxed exponential clock models, paired with constant, exponential and bayesian skyline population models. Markov chain Monte Carlo (MCMC) generations for each analysis were conducted in triplicate for 100 million steps, sampling every 1000 to ensure convergence. Log-combiner was used to collate the log files with a 10% burn-in and Tracer v1.7.2 was used to assess the BEAST runs. The best fitting model was determined by the mean tree likelihood and was found to be the uncorrelated relaxed exponential clock with the Bayesian skyline population model (Supplementary Data 6). TreeAnnotator was used to generate the maximum clade credibility (MCC) tree with a 0.5 posterior probability limit.

### Meta-analysis of CREC isolates

To identify studies reporting carbapenem-resistant *E. coli* we searched PubMed with the search term 'carbapenem* resistant E. coli sequence type*'. The inclusion of studies required publication between 1st of January 2020 to the 27th of March 2025 and MLST analysis on a dataset of ≥50 isolates. These criteria resulted in identification of 33 publications that encompassed 6,877 carbapenem *E. coli* isolates (Supplementary Data 7).

### Statistics and reproducibility

Statistical analyses of bacterial loads in the urine, bladder and kidney of infected mice were performed in GraphPad Prism (version 10.2.2). Other statistical analyses to determine significance for proportion of UTI isolates and carriage of virulence factors were performed in R. All information on statistical tests and sample sizes are described in the appropriate figure legends and methods sections. No statistical method was used to determine sample size. No data was excluded from the analyses. The investigators were not blinded to allocation during experiments and assessment.

### Reporting summary

Further information on research design is available in the Nature Portfolio Reporting Summary linked to this article.

## Data availability

All data generated in this work is presented in the manuscript. Source data are provided with this paper.

## Code availability

The code used in this manuscript to generate figures presented have been made available in the following repository - https://doi.org/10.5281/zenodo.16756974.

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

## Acknowledgements

This work was supported by an Australian National Health and Medical Research Council (NHMRC) Ideas grant to MAS, M-DP and NTKN (2001431) and an NHMRC Ideas grant to MAS and M-DP (2037698). BMF is supported by a University of Queensland Health Research Accelerator grant. We thank the Centers for Disease Control and Prevention (CDC) National Healthcare Safety Network (NHSN), as well as other surveillance networks, for making genome sequence data used in this study publicly available.

## Author contributions

Performed the experiments: L.L.W., B.P., Z.J.L., N.T.K.N., T.C., M.-D.P., K.M.P., K.A.R., C.S. Conceived and designed the experiments: L.L.W., M.-D.P., P.N.A.H., D.L.P., M.J.S., B.M.F., M.A.S. Analysed the data: L.L.W., M.-D.P., B.P., K.A.R., Z.J.L., N.T.K.N., B.M.F., M.A.S. Supervised aspects of the project: M.-D.P., N.P., T.J.K., P.N.A.H., D.L.P., M.J.S., B.M.F., M.A.S. Wrote the manuscript: L.L.W., M.-D.P., B.M.F., M.A.S. All authors read and approved the final manuscript.

## Competing interests

The authors declare that they have no conflict of interest.
