## [Transparent Peer Review file · Nature Communications]

Emergence of a carbapenem-resistant atypical uropathogenic *Escherichia coli* clone as an increasing cause of urinary tract infection

Corresponding Author: Professor Mark Schembri

Version 0:

Reviewer comments:

Reviewer #1

(Remarks to the Author)

I want to thank the authors for taking on board the comments about the limitations of using Enterobase.

This has clearly been taken seriously and they have addressed this (my only major concern) adequately by both altering the language throughout the manuscript, additional analyses of provenance of the strains with available data, and explicit discussion of the limitation in the main text.

Although there are still some uses of the term emergent/ce, but I think the additional work they have done on the source of the isolates with appropriate couching of the findings is now appropriate.

Reviewer #2

(Remarks to the Author)

The manuscript from Walker et al. entitled "Emergence of a carbapenem-resistant atypical uropathogenic *Escherichia coli* clone as an increasing cause of urinary tract infection" is a thorough and well-constructed analysis of an emerging cluster of uropathogenic *Escherichia coli* (UPEC) strains. This manuscript is a revision of a previously submitted paper, and the authors have expanded significantly on their previous work. This manuscript advances our understanding of UPEC genomics and identifies a novel cluster of strains that should be investigated given their relatively rapid emergence and antibiotic resistance profile. This manuscript address an area of concern for human health using a robust suite of in silico tools augmented with judicious use of in vitro and in vivo testing. Given the importance of the subject matter, the strong experimental procedures, and the utility of the research to the microbiology field at large, this manuscript should strongly be considered for publication in Nature Communications.

In their revisions to the manuscript, the authors have addressed the reviewer criticisms thoroughly in all the aspects of the project that were within the scope of this manuscript. The most serious concern for the initial submission was the lack of information on the isolates collected from Enterobase, which the Authors mined for the bulk of their analyses, and the possibility of bias in the submission of genomic information to this dataset. The Authors addressed this concern by providing additional metadata on the on the collection of strains and contextualized their use of Enterobase with existing literature as well as examination of another database. The Authors also addressed issues of overinterpretation in their epidemiological data to enable a more reasonable set of conclusions and provided additional information on the expansion of ST167 over time using additional analyses. Finally, the authors augmented their in silico analyses with in vitro measurements of antibiotic resistance and in vivo measurements of infectivity.

This Reviewer has no major concerns about this manuscript.

Reviewer #3

(Remarks to the Author)

The revised manuscript, "Emergence of a carbapenem-resistant atypical uropathogenic *Escherichia coli* clone as an increasing cause of urinary tract infection," contains a number of improvements and additional analyses beyond the previous version, including testing two strains of ST167 in a mouse model of UTI, a comparison of the prevalence of known virulence factors in ST167 against those in an established UPEC clone (ST131), a meta-analysis of published reports of *E. coli* STs and a Bayesian phylo analysis. The authors have changed the language used to describe the epidemiological trends and added an excellent discussion of the limitations and caveats of using Enterobase data.

I very much appreciate the authors' efforts to carefully address the reviewer concerns, and willingness to add new analyses, which in my view have greatly enhanced the paper. My primary concern with the prior version was related to the potential biases in the underlying genome data, and whether this was impacting the trends the authors were reporting. The new version of the manuscript goes some way to addressing these concerns. I do however, still have some reservations about the evidence that ST167 is specifically emerging as an atypical UPEC. This claim stems primarily from the initial observation of increasing counts of ST167 UTI isolates in Enterobase and then a cross-ST comparison of the proportion of isolates from UTI vs other sources, which shows the proportion for ST167 is not significantly greater than the established UPEC clone, ST131.

i) while the authors have done a great job to change the language and discuss the caveats of the data, I remain nervous about the conclusions. The authors have noted that a large proportion of the data are derived from surveillance programs in the United States- which is good in the sense that these data will represent a more systematic sample, but in the context of the presented phylogeography and the breakdown of isolate sources per sub-clade, I wonder if the apparent association of ST167 with UTI is actually driven by capturing an 'outbreak' of clade C2 associated with UTI in the United States, rather than a global phenomenon that applies to the whole ST? (clade C2 is contributing the highest proportion of UTI isolates and is almost exclusively from the United States – with very short branch lengths in the tree. Other clades contribute a much lower proportion of UTI isolates). Have the authors explored the proportions of UTI vs other sources per ST for subsets of the data stratified by geography and/or subclade (sample sizes permitting)? Do the numbers support more widespread association of ST167 with UTI (even if the sample sizes are insufficient for statistical significance)? If not, perhaps the manuscript title, abstract and conclusions could be clarified to explicitly implicate that the C2 clade may be an emerging UPEC clone of concern, but is so far only detected in the United States?

ii) It is stated that the proportion of isolates that are from UTI for ST167 is not statistically different from the proportions in STs ST131, ST410, ST405, ST361, ST648 and ST1193. Are all of these STs considered major UPEC clones? Are any of the other STs (with lower proportion from UTI) considered UPEC clones? These answers will help to clarify if the trends seen in the data can be interpreted as the authors have implied.

iii) I was very excited to see the inclusion of a meta-analysis to support the authors' findings, but as it stands it doesn't really add much to support the major claims in the paper. It seems that most of the studies in the meta-analysis are not explicitly focussed on UTI isolates- so it doesn't help to support the conclusions on uropathogenicity. Plus, the authors don't dig into the data in any depth. They state: "This analysis from 33 publications revealed that ST167 was the third most frequently detected sequence type across these studies (behind ST131 and ST410), underscoring the clinical relevance and expanding risk posed by this clone (Supplementary Data 7)" But looking at the underlying data in the supplementary table it's clear that not all studies were focussed on clinical isolates and there is substantial variation in the proportion of strains that are ST167 between studies (and geographies?). Since the sample sizes per study were widely varied, it doesn't really make much sense to use the aggregate rank as supporting evidence here. There may well be some useful evidence in the published data, but the current synthesis doesn't demonstrate this.

iv) The mouse model data suggests that the ST167 strains are less fit in the UTI model than the comparator UPEC strain (?), and ST167 carry many fewer UTI-associated virulence factors than the comparator ST131 UPEC clone. The authors conclude that ST167 is therefore an atypical UPEC clone, but I don't see how these data lead to this conclusion without the context of the genomic analyses, and as described above I still have reservations about this. How would any other *E. coli* be expected to perform in the UTI model? And how common are these virulence factors in the broader *E. coli* population? (perhaps the authors can point to evidence in the literature here) Is there any evidence from the mouse model/ virulence factor data that suggest ST167 is more likely to cause UTI than any other *E. coli*?

Reviewer Expertise:

Referee #1: genomic epidemiology, phylogenetic analysis

Referee #2: UPEC, UTIs, genomics

Referee #3: genomics, epidemiology, antibiotic resistance

Reviewers Comments:

Reviewer #1 (Remarks to the Author):

This is an immense amount of analysis that has been done to a good standard, and the figures are well presented. However, it is near-wholly descriptive and is let down by a lack of depth in the contextualisation of other relevant literature, and, critically, an erroneous assumption that the data deposited in Enterobase are representative.

- There is little introduction (just one paragraph) to orient the reader to the space. Additional context e.g. phylogroups, the history of ST131 emergence, current knowledge of AMR distribution and concerns etc would be useful.

Our manuscript was originally submitted as a brief communication with a limit of 2000 words, which incorporated a reduced introduction. In our revised manuscript, we have expanded the introduction to provide additional context to recently emerged antibiotic resistant UPEC clones as suggested by the reviewer (see below).

Lines 62 - 103

Urinary tract infections (UTIs) are one of the most common bacterial infections worldwide, with an estimated ~404 million cases per year and an annual economic burden of US\$3.5 billion in the United States alone^{1,2}. UTIs include infection of the bladder (cystitis), kidneys (pyelonephritis), and can lead to bacterial dissemination into the bloodstream, resulting in life-threatening urosepsis². While antibiotics remain the standard mode of treatment, their widespread use has driven the emergence of resistant strains, particularly among uropathogenic *E. coli* (UPEC), the causative agent of ~70% of UTIs worldwide^{2,3}. Antibiotic resistance to first-line antibiotics such as trimethoprim, co-trimoxazole, ciprofloxacin, and nitrofurantoin, as well as third-generation cephalosporins and last-line carbapenems is of particular concern⁴.

E. coli can be classified into eight broad phylogroups⁵, with phylogroups B2 and D predominantly associated with extra-intestinal infections that include UTI⁶. *E. coli* strains can be further differentiated at higher resolution using multi-locus sequence typing (MLST)⁷. This scheme has identified key UPEC sequence types (STs; also referred to as clones), including ST131, ST1193, ST95, ST73 and ST69, which are responsible for a large proportion of UTIs and bloodstream infections globally⁸⁻¹⁰. These UPEC clones exhibit differences in their antibiotic resistance profiles, with ST131 and ST1193 exhibiting extensive resistance to multiple antibiotic classes⁸. The global emergence and dissemination of ST131 exemplifies the clinical impact of these clones¹¹. ST131 possesses high levels of resistance to fluoroquinolones and third-generation cephalosporins and has become the most common cause of Gram-negative extra-intestinal infections since the turn of the 21st century^{10,11}.

In contrast to these UPEC lineages, ST10 and the ST10 clonal complex (ST10cc) belong to *E. coli* phylogroup A, a group typically associated with intestinal commensal colonisation⁵. The ST10cc includes STs such as ST10, ST167, ST617 and several other STs¹². While *E. coli* strains from the ST10cc are not typically considered a major cause of extra-intestinal infections, there are reports of STs found

in the ST10cc that exhibit high rates of carbapenem resistance and cause UTI and bloodstream infections^{10,13,14}.

In this study, we analysed the Enterobase¹⁵ dataset and identified an increasing number of ST167 genomes from UTI/urine samples between 2020-2024. Phylogenetic analysis of a subset of 800 ST167 genomes defined the clade structure of this high-risk clone and identified an emergent subclade (which we refer to as subclade C2) with very high carriage (89%) of genes encoding resistance to carbapenems and cephalosporins. In silico serotyping of ST167 genomic data revealed the acquisition of two group 1 capsule types within subclade C2, namely KL124^{Group1} (KL124^{G1}) and KL30^{Group1} (KL30^{G1}). We also found that ST167 strains carry significantly reduced numbers of UPEC virulence genes compared to ST131 strains, including an absence of genes encoding type 1 fimbriae in the majority of ST167 strains. Together, our analyses provide insight into the evolution of this emerging uropathogenic clone and identify distinct genomic signatures that support the classification of ST167 as an atypical UPEC clone associated with UTI.

- The premise of the paper from Line 60 – 85 that ‘ST167’ is emerging is not supported. A variety of epidemiological language is inappropriately used as equivocal to an increase in database depositions throughout e.g. emergent, striking increase in prevalence, recent expansion’ etc. This is effectively treating what is in Enterobase as surveillance data (i.e. systematically collected in a non-ad hoc manner), which it is not. It is public data, and the paper presents no effort to try and convince the reader otherwise (e.g. strains coming from public health agencies, or engagement with the data generators from major projects).

We agree with the reviewer that Enterobase is not a surveillance database and have taken this criticism onboard in our revised manuscript (see below and responses to other reviewer comments). We have also toned down the epidemiological language in line with the reviewer’s comments. While we agree that the data in Enterobase is not surveillance data, we also note that Enterobase should not be seen as a singular, standalone repository any more than the Sequence Read Archive (SRA). The vast majority (if not all) of data generated from sequencing projects is deposited in either the SRA, European Nucleotide Archive (ENA) or DNA Data Bank of Japan (DDBJ). If these data meet specific quality control criteria, they naturally flow into platforms such as Enterobase. This means that while Enterobase may not cover every nuance of epidemiological data, its main bias is essentially in its taxonomic inclusion, which reflects underlying sampling from the SRA rather than a design flaw in Enterobase itself.

We acknowledge that Enterobase will be subject to influence based on research studies and the regions in which they take place. We do not suggest that ST167 is a larger cause of urinary tract infections compared to ST131 or any other sequence type for that matter. In our revised manuscript, we have modified the way we present this interpretation (see below).

We have also included a discussion on the limitations of using public data available in Enterobase in the new concluding remarks section.

Enterobase analysis - Lines 107-122:

We screened Enterobase¹⁵ to detect signals of emerging *E. coli* lineages and found distinct patterns in the proportion of *E. coli* sequence types (STs) that cause UTIs. Data from 2024 (database accessed October 4th, 2024) revealed that ST167 represented the highest number of sequenced isolates from human UTI/urine (96 genomes), followed by ST410¹⁶ (61 genomes) and ST131¹¹ (50 genomes) (Supplementary Fig. 1; Supplementary Table 1). The two most frequently sequenced STs from all isolate sources were ST10 and ST167¹⁷, both of which belong to phylogroup A and the ST10cc¹². Strains from *E. coli* phylogroup A are typically associated with intestinal commensal colonisation⁵. ST10

showed minimal association with UTI/urine (4/371 isolates). In contrast, almost half of ST167 isolates were from UTI/urine sources (96/197 isolates), higher than the pandemic UPEC ST131 clone (50/136 isolates) (Fig. 1a). Longitudinal analysis of UTI/urine isolates in EnteroBase from the five most deposited STs revealed a recent increase of ST167. ST167 increased ~17-fold in the EnteroBase dataset, from 1.2% of total UTI/urine isolates in 2020 to 21.2% in 2024 (Fig. 1a; Supplementary Table 2). The increase in ST167 depositions to EnteroBase, coupled with the high number of sequenced isolates from UTI/urine sources, suggests this ST represents an emergent phylogroup A uropathogenic clone.

Limitation section – Lines 357-387:

This study had several limitations. The genomic data was sourced from EnteroBase, a publicly available database that primarily obtains its data from the NCBI Short Read Archive (SRA). Although EnteroBase offers valuable insights into genomic epidemiology, its use of publicly submitted sequencing data reflects the sequencing activities and priorities of contributing researchers rather than a structured surveillance system. Consequently, the data does not provide a comprehensive representation of all bacterial populations or geographic regions, which may affect the generalizability of the study's findings. We note that the majority of North American ST167 isolates (89.5%) were collected as part of the Centers for Diseases Control and Prevention (CDC) National Healthcare Safety Network (NHSN). The NHSN is a large program that conducts surveillance for healthcare events across diverse facilities in the United States, including the collection of data and biological samples associated with healthcare-associated infections (HAIs) and antimicrobial resistance⁴⁵. Leveraging the NHSN extensive surveillance data, we inferred an increase in the proportion of genome sequenced UTI-associated *E. coli* ST167 in North America. However, the high number of genomes contributed to EnteroBase through this program means there are sampling biases in the data, for example the overrepresentation of human pathogens that: (i) are sequenced from high resource settings, (ii) cause HAIs, and (iii) possess resistance to critical antibiotics such as carbapenems. Thus, while our findings identify ST167 as a frequently sequenced UPEC clone that causes UTI, we do not have sufficient data to compare infection incidence to other global clones such as ST131. The geographical pattern we observed reflecting an increasing number of ST167 genomes with carbapenem resistance isolated in North America is supported by the NCBI pathogen detection tool, where SNP cluster PDS000063179.375 (n=528 isolates), encompassing HC20 cluster 109409 of subclade C2, shows 87.6% of isolates originate from this region (Supplementary Fig. 7). Despite this, the assessment of subclade C2 in other regions is limited by the availability and scope of comparable surveillance data. To further address potential database-driven biases, we conducted a meta-analysis of published studies from 2020 to 2025 that reported MLST data on carbapenem-resistant *E. coli* isolates, with a threshold of at least 50 isolates required for inclusion. This analysis from 33 publications revealed that ST167 was the third most frequently detected sequence type across these studies (behind ST131 and ST410), underscoring the clinical relevance and expanding risk posed by this clone (Supplementary Data 7).

How do they know this is not a focused research study in a region where this ST is endemic, or one focused on the phylogroup that is responsible for the increase?

We appreciate the reviewer's feedback and concern relating to our conclusion around the increasing proportions of ST167 genomes from UTI/urine sources. The *E. coli* genomes from UTI/urine sources used between 2010-2024 were sourced from 77 sequencing projects (Bioprojects) and collectively span 6 continental regions (North America ~61.7%, Europe ~14.6%, Asia ~13.8%, Africa ~4.5%, South America ~1.9%, Oceania ~1.8%). Data sources, including temporal and spatial metadata have been included in new Supplementary Data 8.

It is clear from these data that isolates from North America constitute a significant proportion (61.7%) of study isolates. However, the majority of North America ST167 (89.5%) were collected as part of the Centers for Diseases Control and Prevention (CDC) National Healthcare Safety Network (NHSN). The NHSN is a large program that conducts surveillance for healthcare events across diverse facilities in the United States, including the collection of data and biological samples associated with healthcare-associated infections (HAIs) and antimicrobial resistance. Specifically, the ST167 C2 genomes from the CDC dataset (BioProject PRJNA288601) represent isolates from 25 states. Given the broad scope of the NHSN, we used the data to infer an increase in the proportion of UTI-associated ST167 in North America. However, we acknowledge that the availability and scope of corresponding surveillance data constrain the assessment of subclade C2 in other regions. In our revised manuscript, we have presented more detail of the Bioprojects and added new meta-analysis sourced from 33 publications in the period 2020-2025 that reported ST distribution associated with carbapenem resistance (see response to reviewer 2; new Supplementary Data 7).

We have now included discussion of this topic in the limitations section (see above).

- The major phylogenetic analysis appears methodologically fine but is wholly descriptive and again makes comparisons about the relative proportions of sample provenance in clades which don't really say a lot in the face of the unknown provenance of the isolates. In any event, the values presented at L97 – 99 need exact numbers presented alongside the statistical significance measures.

We acknowledge the limitations pointed out by the reviewer. As mentioned above, we have now included a section in the discussion that addresses these limitations.

With respect to the values referred to by the reviewer, we have adjusted the text to include exact numbers alongside the statistical significance.

The following modification has been made in the revised manuscript to address this comment.

Lines 153-157:

Further analysis of the isolation source revealed that subclade C2 had a significantly higher association with UTIs and urine samples (109/261 isolates [41.8%]) compared to clade A (10/70 isolates [14.3%]; $P = 6.53 \times 10^{-5}$), clade B (60/314 isolates [19.1%]; $P = 1.91 \times 10^{-8}$) and subclade C1 (7/38 isolates [18.4%]; $P = 4.14 \times 10^{-2}$; Fisher's exact test with Bonferroni-adjusted p -values) (Fig. 1d).

Regarding, the unknown provenance of the isolates, we believe that by comparing the clades in this way provides valuable insights including the year of isolation (and year range used in Fig 1), association with extra-intestinal infection and antibiotic resistance (presented in Supplementary Data 2). In line with the reviewers comments, we agree that additional metadata would enhance the insights into ST167 infection. However, we do not have access to such data.

- Line 163 – there are tools available for E. coli capsular typing e.g. <https://journals.asm.org/doi/10.1128/jcm.00008-15>

We thank the reviewer for alerting us to the SeroTypeFinder tool. This tool is primarily used for O-antigen and H-antigen typing. However, some group 1 and group 4 capsules are given an O designation by this tool, enabling classification of certain K antigens, e.g. K87 is O32, K85 is O141, K9 is O104.

Lines 293-295:

Given the high levels of similarity between the group 1 capsule of *E. coli* and *Klebsiella pneumoniae*³⁶, we employed Kaptive 2.0²², a *K. pneumoniae* capsule typing tool.

- Portion C – again – inappropriate use of prevalence – Proportion resistant

We have altered the text throughout and now refer to ‘proportion resistant’ as suggested.

- Why not explore the *papGII* locus if arguing that this presents a new and emerging UPEC threat?

ST167 subclade C2 does not contain the *papGII* locus. Only a small proportion of ST167 clade B contain *pap* fimbriae genes (See Figure 2a).

Reviewer #2 (Remarks to the Author):

Synopsis:

The manuscript “Emergence of carbapenem-resistant *Escherichia coli* ST167 as a significant cause of urinary tract infection” by Walker et al. is a description of an emerging lineage of uropathogenic *Escherichia coli* (UPEC). The authors provide a detailed and well-structured investigation into the genomic features of ST167 and its population structure, which includes subclades A, B, C1, and C2. Particular focus is given to subclade C2 which is enriched in UTI isolates and carries resistance genes against last-line antibiotics. The emergence of this uropathogenic lineage is interesting given the fact that it is housed within phylogroup A in UTI, even though these phylogroup A strains are not frequently associated with UTI. The authors describe the rapid emergence of subclade C2 in the last 4-5 years and describe important genomic features of pathogenicity, including changes to O-antigen and flagella. The authors provide convincing data on the rapid emergence of a unique uropathogenic lineage of *E. coli*, which is somewhat limited by their reliance on a single database for genomic information (Enterobase). The authors are careful, transparent, and thorough in their use of genomic tools to define the population features of ST167 and their figures and data are presented well and clearly. There are limitations to the manuscript, mainly in the lack of any exploration of the ST167 uropathogenicity in the available mouse models of UTI, which the corresponding authors have experience. Further, additional comparative genomic analyses to provide context as how ST167 genomes are unique (especially subclade C2) would be interesting and pertinent. Overall, I believe that the manuscript is well-constructed, but it is missing some key elements (namely measures of urovirulence and more comparative genomics). As such, I do not believe that it warrants publication in *Nature Microbiology* in its current state; however, if these critiques (described in more detail below) are addressed, then I believe that this manuscript could warrant publication in *Nature Micro*.

Comments of major concern:

-Genomic comparisons between the closely related A, B, and C subclades, especially subclade C2, could highlight genomic features associated with human UTI. In general, great context about genomic features that differentiate ST167 from other phylogroup A strains and differentiate ST167 subclades would be appreciated. The comparison with phylogroup A is particularly interesting given the fact that so few A strains are uropathogenic.

We thank the reviewer for this question and agree that this would provide greater context on the genomic features of ST167. To address this question, we curated a list of 74 non-redundant marker genes for established UPEC virulence factors to highlight features associated with human UTI (new Supplementary Data 4). We first performed a comparative analysis for carriage of UPEC virulence

factors with ST131 (new Supplementary Figure 4). This analysis revealed a reduced number of virulence factors for ST167. We expanded this analysis to assess the UPEC virulence factors found across the ST167 phylogenetic structure and showed clade-specific patterns (new Figure 2). This analysis can be read in greater detail below as a **new section** in our updated manuscript.

Regarding the broader comparison with phylogroup A strains, we agree this is an interesting angle given the typically non-uropathogenic nature of phylogroup A. However, given the extensive genetic diversity within phylogroup A and the substantial analytical work such a comparison would require, we argue this represents a separate research question that extends beyond the scope of the current manuscript.

Lines 205 - 257

To investigate the virulence potential of ST167, we conducted a comprehensive analysis of UPEC virulence genes across 800 genomes from our phylogenetic dataset. We curated a list of 74 non-redundant marker genes for established UPEC virulence factors, encompassing genes involved in adhesion, iron acquisition, immune evasion, motility and toxin production (Supplementary data 4). Comparative genomic analyses between ST167 and ST131 revealed that ST167 had significantly fewer UPEC virulence factors corresponding to chaperone-usher fimbriae, autotransporters, iron uptake, toxins and immune evasion mechanisms (Supplementary Fig. 4). The distribution of virulence factors within the phylogenetic structure of ST167 also revealed clade-specific patterns; for example, the presence of a specific *agn43* (antigen 43) allele in subclade C2 and absence of the yersiniabactin receptor gene *fyuA* in subclade C2 (Fig. 2a). Analysis of chaperone-usher fimbriae revealed the presence of genes encoding Ecp/Mat fimbriae (*ecpC*), Yad fimbriae (*yadV*) and Yfc fimbriae (*yfcV*) in the majority of strains, and Yeh fimbriae (*yehC*) in strains from clade B, C1 and C2. Strikingly, all ST167 strains except those from clade A lacked type 1 fimbriae (*fimD*). Further investigation into the absence of type 1 fimbriae genes revealed a complete replacement of the entire *fim* locus by the IS1A insertion element (Fig. 2b). Type 1 fimbriae are a critical UPEC virulence factor required for bladder colonisation^{24,25}. Type 1 fimbriae-mediated binding via the tip-located FimH adhesin facilitates UPEC adhesion to α -D-mannosylated glycoproteins such as uroplakins on the bladder epithelium²⁶, facilitating invasion of bladder superficial epithelial cells²⁷ and the formation of intracellular bacterial communities (IBCs)²⁸. To assess whether ST167 subclade C2 could colonise the bladder despite the lack of type 1 fimbriae, we employed a murine UTI infection model. ST167 subclade C2 strains MS25298 and MS25303 were recovered at variable loads in the urine, with bacteria detected in 11/19 and 5/19 mice, respectively. In contrast, the reference UPEC strain CFT073 was recovered in significantly higher bacterial loads in the urine (Fig. 2c). Bladder colonisation of MS25298 and MS25303 were also significantly reduced (median = 0) compared to CFT073 (median = 10^7 CFU/g), with bacteria detected in 1/19 and 7/19 mice at approximately 10^4 CFU/g, respectively (Fig. 2d). None of the strains effectively colonised the kidneys (Fig. 2e), consistent with previous data from our lab using CFT073 for infection of C57BL/6 mice^{29,30}. Taken together, our discovery that type 1 fimbriae are absent in most ST167 leads us to propose ST167 as an atypical UPEC clone associated with UTI.

Supplementary Figure 4. Comparative carriage of virulence genes in ST131 and ST167. The 2,066 available ST167 genomes and a previously published ST131 dataset (3,993 genomes) were screened by ABRicate to determine the carriage of virulence factor marker genes. The analysis used a curated dataset of established UPEC virulence factor marker genes (Supplementary Data 4). To assess a statistically significant difference in proportion of carriage, a Chi-squared test with Yates correction. The Bonferroni correction was performed to adjust for multiple testings, with significance levels indicated by asterisks: *** $p < 0.001$, **** $p < 0.0001$, after Bonferroni correction. Colouring of asterisks red indicates higher proportion in ST167 and colouring in blue indicates higher proportion in ST131.

Figure 2. a, Virulence factor heatmap of ST167. On the left is the phylogenetic tree of ST167 depicting the 800 genomes shown in Fig. 1a. The heatmap represents the presence and absence of the virulence factor marker genes, black represents presence. **b**, An EasyFig³¹ comparison of the type 1 fimbriae locus in CFT073 and representative complete genomes from clades A, B, C1 and C2. **c-e**, Murine UTI model showing bacterial loads in urine, the bladder and the kidneys. Each symbol represents data from an individual mouse at 24 h post infection. MS25298 and MS25303 are subclade C2 genomes from HC20 cluster 109409 and contain the KL124 capsule. Statistical significance was determined by

an ordinary one-way ANOVA with the Dunnett correction applied; ST167 strains were compared to CFT073, significance levels are indicated by asterisks: *** $p < 0.001$, **** $p < 0.0001$.

-The interpretation of the data presented here are limited by the use of a single genomic database for the vast majority of the data presented (Enterobase). If the authors are able to replicate their findings in other databases, this would solve this critique. If that is not possible, then the authors should provide additional context to their work over 2010 to 2024 (Supplementary Figure 1B) to ensure that equivalent numbers of distinct studies, geographical regions, and (when possible) total genomes uploaded are similar over time. Each of these (different research groups, countries/locals, and number of genome uploads) could bias the representation of human-associated *E. coli*. These data could be compiled as another Supplementary Table to give the readers additional context.

We thank the reviewer for this comment and suggestion. As requested, we have generated a new Supplementary Table that includes temporal and spatial metadata from the data sources of contributing sequencing projects (new Supplementary Data 8). As presented in our response to reviewer 1, Enterobase incorporates data generated from sequencing projects deposited in either the SRA, European Nucleotide Archive (ENA) or DNA Data Bank of Japan (DDBJ), and thus should not be seen as a singular database. However, to support our findings from Enterobase and address the reviewers concern, we performed a meta-analysis of all studies that conducted multi-locus sequence-typing of carbapenem-resistant *E. coli* with more than 50 isolates, published between 2020-2025. This meta-analysis has been included as a Supplementary Table in our updated manuscript (new Supplementary Data 7). The meta-analysis collated 33 studies that included 6,877 carbapenem-resistant *E. coli* isolates. From our analysis, ST167 was the third-most isolated sequence-type, behind ST131 and ST410. These studies provide additional support to our finding that ST167 with carbapenem-resistance isolated from healthcare-associated infection in North America by the CDC is of clinical risk (see response to reviewer one).

-Line 230: The sentence “The combined acquisition of a new group 1 capsule and extensive antibiotic resistance may be driving factors in the recent expansion of ST167 subclade C2” is an overinterpretation of the data presented. While the authors have been convincing ST167 clade C2 is enriched in certain genomic features, the authors have not provided evidence about which genomic features are drivers and which features are “passengers” in the expansion of ST167 clade C2.

We agree. In the revised manuscript we have removed the claim suggesting the combined acquisition of a new group 1 capsule and extensive antibiotic resistance may be driving factors in the recent expansion of ST167 subclade C2.

-In Figure 1a, please use colorblind friendly colors. The colors for Rectal and Blood ticks for the phylogenetic tree are particularly difficult to differentiate. If possible, adding gaps between the color wheels would also be appreciated (e.g., a thin white line between “year” and “source”)

We thank the reviewer for alerting us to this issue and have adjusted our manuscript as suggested. Furthermore, we have avoided using red and green within the same columns to make the figures colour blind friendly. See Figure 1 below.

Figure 1. a, Bar graph displaying the human UTI/urine isolate proportion (%) from 2010 to 2024 for the five most sequenced *E. coli* STs deposited to EnteroBase in 2024 (source data in Supplementary Table 2). **b**, A Maximum likelihood tree of 800 ST167 genomes. The tree was constructed from 109,894 recombination-free core-genome polymorphic sites by IQ-TREE²⁰ (GTR+G model and 1000 ultrafast bootstraps²¹; with CP083869.1 used as reference and ED1a as outgroup). Tree scale is representative of nucleotide substitutions per site. ST167 clades A-B and subclades C1-C2 are shaded and were determined using fastBAPS¹⁹. The HC20 clusters are coloured on tip points of the branches. The pink stars on the branches of the major clade are indicative of $\geq 90\%$ bootstrap support. Kaptive 2.0²² was employed to determine the capsule type of the ST167 genomes. AMRfinderplus²³ was used to

determine the resistance genes carried by the ST167 genomes, with red on the outermost ring indicating carriage of both carbapenem and cephalosporin resistance genes. **c-f**, EnteroBase¹⁵ metadata was compiled to display bar charts for (c) collection year, (d) isolate source, 10/70 isolates were from UTI/urine sources for clade A, 60/314 for clade B, 7/38 for subclade C1 and 109/261 for C2, (e) carbapenem and cephalosporin resistance, and (f) capsule type.

-Why were subclades required to include more than 30 genomes? What is the reasoning behind that number? An explanation, with data support, is warranted here. Maybe the authors could provide an estimation of the average and maximum phylogenetic distances of clades with 30 genomes from Figure 1, as compared to clades with 20 genomes, etc?

We appreciate the reviewers concern regarding the 30 genomes threshold and in our updated manuscript have removed this threshold. The reviewer is correct to question why a 30-genome threshold was implemented to define clades/subclades. In our initial submission we used a 30-genome threshold as an arbitrarily chosen number. If we were to be strict, a cluster identified by FastBAPS would be considered a clade in the phylogenetic tree. In our new manuscript we have recognised the clusters identified at all levels by FastBAPS (see below), however, to make this easier to follow we have provided clading nomenclature to the largest clades: A, B and C. Given the small size of the remaining clusters (1-23 genomes), we have not provided a formal clade name to these clusters in this manuscript. If they were to continue to expand in the future, then subsequent work could build on our analysis and formally provide these clusters with a clade name. We have provided information regarding FastBAPS clusters in new Supplementary Data 1.

Lines 145 - 148:

Bayesian analysis of population structure (fastBAPS)¹⁹ identified 17 distinct clusters at BAPS level 1 and level 2 (Supplementary Fig. 3). The three largest clusters were defined as clade A (70 genomes), B (314 genomes) and C (329 genomes), with the 14 remaining clusters ranging in sizes from 1-23 genomes (Supplementary Fig. 3; Supplementary Data 1).

Supplementary Figure 3:

Supplementary Figure 3. Bayesian analysis of population structure by fastBAPS¹⁹ at three sequential levels. Clade A was defined at BAPS level 1 (blue), clade B and C at BAPS level 2 (purple) and the subclades of C at BAPS level 3 (red).

-Line 145: Why did the authors choose <10 allelic differences as a cut-off? Is this an arbitrary cut-off, and, if so, how does this changing this number alter the authors' measurements?

EnteroBase offers hierarchal clustering cutoffs of 0, 2, 5, 10, 20, 50, 100, 200, 400, 1100, 1500, 2000 and 2350. The <10 allelic differences were chosen arbitrarily as this cutoff provided separation between the two main capsule types of subclade C2. If the cutoff is increased to HC20 or HC50, the majority of subclade C2 forms one cluster and other clades are preserved. Increasing the cutoff to HC100, results in some merging between subclade C1 and B. If we were to decrease the clustering to

HC5 it would lead to further fragmentation of HC10 clusters. Following feedback from reviewer 3 regarding the comparison of HC10 clusters, we have simplified our analysis and now present HC20 clusters that contain more than 100 isolates. This arbitrary cutoff allows us to perform a pairwise comparison of HC20 clusters that make-up at least ~5% of the total isolates. The largest HC20 clusters in ST167 (n = 2,066 genomes) are 109409, 64736 and 104015 that each contained 432, 173 and 78 genomes, respectively. We focussed on the HC20 clusters 109409 and 64736 as these were the most prominent in our dataset.

-Have the authors considered testing some representative subclade C2, A, and B strains in a mouse model of acute UTI to determine if the strains are competent for infection? Most model UPEC strains used in murine models of UTI are phylogenetic clade B2, so it would be interesting to see if ST167 strains follow a similar course of infection as model B2 strains. Excitingly, maybe the ST167 strains colonize in a different manner!

We thank the reviewer for the excellent suggestion. In response, we have performed mouse UTI experiments using two representative subclade C2 strains (from the HC20 cluster 109409) together with the UPEC reference strain CFT073 as a control (New Figure 2). See also response to comment 1.

We agree that extending this analysis to include strains from clades A and B would provide valuable comparative data. As of yet, we have not sourced clade A or C1 strains for the current study but plan to in future work. We believe that for the current manuscript, experimental data with subclade C2 strains provides a significant advance on our current knowledge of ST167 pathogenesis.

-Additional test of ST167 strains for their phenotypic differences in antibiotic resistance is also warranted, especially in connection with the breadth of their predicted resistance profile. Perhaps there are additional resistances that have not been identified *in silico*.

We agree. As indicated above, we acquired ST167 isolates from the dominant HC20 cluster 109409 (clade C2; isolates sourced from Germany) and from clade B (isolates sourced from Australia). We performed antibiogram testing of these isolates and these results largely correlated with our *in silico* predictions (new Supplementary Data 3).

Lines 166 - 172:

To validate the *in silico* resistance profiles, we sourced strains from clade B (n = 6) and subclade C2 (n = 5) and conducted susceptibility testing for 24 antibiotics across 9 different classes. Most isolates from clade B and C2 were non-susceptible to cephalosporins and carbapenems, while there were differences in susceptibility for the isolates examined to amikacin, gentamicin and minocycline (Supplementary Data 3). All isolates examined were susceptible to nitrofurantoin (Supplementary Data 3).

-The authors state that the O89 antigen may influence ST167 pathogenicity. This is a testable hypothesis that warrants exploration in a mouse model of infection. If a clade ST167 clade C2 strain is not available, could the authors try taking another phylogroup A strains and exchanging the O antigen to O89?

We agree this is a valuable experiment, however, we believe it is beyond the scope of the present study. The reviewer has suggested an alternative approach to swap the O-antigen. A method for this approach was recently published for *Pseudomonas aeruginosa* (PMID: 39621751), however swapping the O-antigen genes in UPEC has not been performed previously. The O-antigen locus is large, and thus methods for performing a specific gene swap are complex, complicated further by

the fact that we also need to modify our genetic tools to enable the construction of mutants using the Lambda Red recombinase system (the ST167 strains are resistant to the antibiotics we use for other UPEC strains, including ST131 strains). Together, we believe such a large body of extra work is beyond the scope of the present study.

Comments of minor concern:

-Line 67: What does the phrase “consistent with its classification in phylogroup A” mean? Please explain.

We agree the statement was ambiguous and have rephrased it in the revised manuscript. Our intention was to highlight the fact that strains from Phylogroup A are primarily considered to be gut commensal *E. coli*. The metadata from Enterobase revealed that 4/371 ST10 isolates from 2024 were associated with a UTI/urine source. Thus, very few isolates from ST10 are sourced from UTI urine, compared to isolates from ST167, which were isolated from UTI/urine sources in greater numbers (96/197 isolates). We have changed our text as follows:

Lines 111 - 117:

The two most frequently sequenced STs from all isolate sources were ST10 and ST167¹⁷, both of which belong to phylogroup A and the ST10cc¹². Strains from *E. coli* phylogroup A are typically associated with intestinal commensal colonisation⁵. ST10 showed minimal association with UTI/urine (4/371 isolates). In contrast, almost half of ST167 isolates were from UTI/urine sources (96/197 isolates), higher than the pandemic UPEC ST131 clone (50/136 isolates) (Fig. 1a).

-Line 67: Does the difference in proportion of UTI strains from different STs reach statistical significance, perhaps via a Fisher’s exact test?

A pairwise Fisher’s exact test for the proportion of UTI isolates from different sequence types does not lead to statistical significance for ST167 compared to ST131, ST410, ST405, ST361, ST648 and ST1193 after correction of multiple comparisons by the Bonferroni correction. ST167 is statistically significant compared to all other tested sequence types. We have included this in the figure legend of Supplementary Figure 1 and new Supplementary Table 3.

Lines 131 – 136:

A pairwise Fisher’s exact test for the proportion of UTI isolates from different sequence types did not lead to statistical significance (p value > 0.05) for ST167 compared to ST131, ST410, ST405, ST361, ST648 and ST1193 after correction of multiple comparisons by the Bonferroni correction. In contrast, the increase in the number of ST167 sequenced isolates was statistically significant (p value < 0.05) compared to ST69, ST38, ST10, ST162, ST58, ST297, ST88, ST349, ST504, ST48, ST3580, ST73 and ST95 (Supplementary Table 3).

-In Fig 2b, there is a typo. Replace “Humans” with “Human”

Replaced as suggested.

Reviewer #3 (Remarks to the Author):

The manuscript, “Emergence of carbapenem-resistant *Escherichia coli* ST167 as a significant cause of urinary tract infection,” describes an analysis of publicly available genome data from which it is inferred that *E. coli* ST167 is an emerging cause of urinary infections, with recent emergence of a

particular subclade (C2) with high rate of cephalosporin and carbapenem resistance. This subclade was primarily represented by isolates originating from the United States, and was distinguished by chromosomal recombination events that result in a change of capsule and H antigen as compared to the most closely related ST167.

Urinary tract infections are associated with a major disease burden globally, and E. coli is the dominant pathogen. Analyses of the major genotypes associated with disease, and detection of novel high risk clones (or subclades thereof) can provide key insights to guide surveillance and intervention strategies, and empiric treatment guidelines. Therefore, the topic of this work should be of interest to a broad audience.

The paper is very well written, easy and enjoyable to read. The genomics approaches match the gold standard and are adequately described. My one major concern stems from the use of public genome collections without any apparent consideration of the sampling biases in the data e.g. overrepresentation of sequences from high resource settings, of human infection causing strains, MDR / carbapenem resistant strains etc. Given these biases, any calculated prevalences or ratios e.g. ratios of isolates from UTI vs other sources, rates of carbapenemase carriage, should be considered very carefully and the potential impacts on the findings carefully explained.

We agree, and as indicated above we have included a new section in our revised manuscript to address the limitations of our study. The points raised above are included in this section as described in our concluding remarks (see response to Reviewer 1).

Similarly, the phylogeographic analyses need to be considered in the context of the available data. For example, the statement that clade C2 has emerged since 2020 in the United States could easily be an artefact of bias in dataset where the true country of origin is not well represented and the clade first reaches a country with high sequencing rates in 2020. The authors should explore and explain the biases in the data, and acknowledge the relevant caveats to their findings.

We agree and have now removed our suggestion that C2 has emerged from North America. As described in our response to Reviewer 1, we have included a section within the concluding remarks that address the limitations of our study. The points raised in this query are included in this section.

In addition, I have a small number of minor comments:

1. Lines 261-264: - are there any specific justifications for the QC criteria used here?

The EnteroBase platform uses QC criteria such as: size 3.7 Mbp – 6.4 Mbp, N50 value >20kb, number of contigs =< 800, proportion of scaffolding placeholders (N's) <3% and species assignment using Kraken >70% contigs are assigned (<https://enterobase.readthedocs.io/en/latest/pipelines/backend-pipeline-qaevaluation.html>).

As we wanted to generate a phylogenetic tree using core genome SNPs, we required high quality genomes with (i) low scaffolding placeholders and (ii) less fragmentation. Thus, in addition to EnteroBase thresholds, we utilised a <0.02% proportion of scaffolding placeholders per genome threshold (in comparison to <3% used by Enterobases) and contigs < 500. This resulted in a higher quality dataset that was applicable for our downstream analyses, including core genome SNP phylogeny.

2. Lines 142-157: the relevance of the comparisons here are unclear. The non-C2 HC10 clusters seems to be a small subclade in the tree that happens to be represented by a high number of

isolates (maybe it's from an outbreak investigation? Or a highly sampled population?) Why focus on this as a comparator and ignore all of the other genomes?

We appreciate the reviewer's query. To reduce the ambiguity of this section we have simplified our analysis to assess the genomes at the HC20 level. By assessing clusters that comprise ~5% of the total isolates (>100 genomes each), we were able to investigate large clusters in clade B and subclade C2. We observed the cluster in clade B (HC20 64736) during our analyses and while this was not a focus of our study, we acknowledge it has previously been described (PMID: 31961309; PMID: 40239363). This cluster has been sequenced by surveillance networks in both the US and China as a cause of veterinary and human infections. By performing a pairwise comparison between HC20 cluster 109409 and 64736, we were able to focus on these emerging lineages and provide information relating to the carriage of clinically relevant AMR genes.

3. Line 223: "Subclade C2 shows evidence of recent clonal expansion, particularly in North America since 2020." – aside from the impact of potential sampling biases mentioned above, this sort of bold statement about 'recent expansion' would be strengthened by some kind of Bayesian population analysis.

We performed a BEAST analysis on subclade C2 as suggested by the reviewer (**new Fig 4**). This analysis demonstrated that following the most recent common ancestor (TMRCA) of subclade C2 there was an increase in effective population size that represents a clonal expansion. This analysis does not come without limitations that we acknowledge in the text. Firstly, we observed a positive temporal signal suggesting a clock-like evolution, however, the root-to-tip regression analysis exhibited an R^2 value of 0.1057. We believe that this analysis was limited due to metadata only describing year-level collection data for all isolates. To maintain a uniform dataset, all isolates were assigned January 1st of the collection year. The implication of this would be that the overall estimated evolutionary change would not be as accurate. Ultimately, these constraints may affect the variability of dating the TMRCA of the HC20 cluster. Despite this, the conclusion that the HC20 cluster of subclade C2 is a major contributor to the clonal expansion of subclade C2 is valid.

Lines 332 – 354:

The geographical distribution of subclade C2 shows that 87.6% of the strains were isolated in North America, associated with the expansion of the HC20 cluster 109409 which composed ~77% of subclade C2 isolates (Fig. 4a). To assess the population dynamics of subclade C2 we extracted all subclade C2 genomes from EnteroBase ($n = 539$) and assessed the temporal signal with TempEST⁴³. There was evidence for a modest clock-like evolution (correlation coefficient = 0.3252; $R^2 = 0.1057$). The identification of the clock-like structure of subclade C2 was constrained by limited metadata detailing collection date, with only year of isolation consistently available across isolates. Despite these constraints, the positive temporal signal allowed us to employ BEAST⁴⁴ to reconstruct the evolutionary timeline of subclade C2. We determined the best-fitting model to be the uncorrelated relaxed exponential clock with the Bayesian skyline population model based upon the mean tree likelihood (Supplementary Data 6). The time to most recent common ancestor (TMRCA) of the HC20 cluster was 2015.8 (95% highest posterior density (HPD): 2014.9 – 2016.5) (Fig. 4a). The expansion of the HC20 cluster 109409 correlated with the Bayesian skyline plot that showed a clonal expansion around 2016 (Fig. 4b). Taken together, these data suggest HC20 cluster 109409 is an expanding clone with two distinct serotypes, primarily sequenced in the United States and associated with UTI.

Figure 4. a, Time scaled tree of ST167 subclade C2 consisting of 539 genomes. The pink polygon depicts the node correlating to the TMRCA of HC20 cluster 109409. Metadata identifying HC20 cluster, capsule type and continent of isolation are displayed on the right as columns. **b,** Bayesian skyline plot of subclade C2 genomes with the dotted line corresponding to the TMRCA of the HC20 cluster 109409.

4. Lines 219-220: “In conclusion, our genomic investigation provides an explanation for the emergence of ST167 as a multidrug resistant UPEC lineage associated with UTI” I disagree that this statement is supported in the data. The genomics analyses have identified distinguishing features of the C2 subclade compared to other ST167 clades, but there is no direct evidence in this work that these features are causing enhanced uropathogenicity – in fact there isn’t even any logical reasoning given as to why this particular capsule and H antigen combination might result in greater virulence than any other combination. Suggest toning down this claim.

We agree, and have toned down our conclusion as follows:

Lines 389 - 403:

In conclusion, our genomic investigation provides insights into the emergence of ST167 as a multidrug resistant UPEC clone associated with UTI, an observation that builds on other reports linking ST167 with increasing carbapenem resistance^{13,46,47}. A striking finding from our genomic investigation of ST167 was the complete loss of genes encoding type 1 fimbriae in almost all ST167 strains from clade B, C1 and C2. Given the critical role of type 1 fimbriae for UPEC bladder colonisation in mouse models that replicate UTI^{24,25}, the inability to produce these fimbriae suggests ST167 disease pathogenesis is different to previously characterised phylogroup B2 and D UPEC strains and associated with alternative mechanisms of colonisation and/or virulence factors. The group 1 capsule identified in ST167, likely originating from *Klebsiella*, may be one such virulence factor. Our discovery also raises important considerations concerning ongoing UTI programs aimed at the development of new anti-virulence drugs to block binding of the type 1 fimbriae FimH tip adhesin to the bladder uroepithelium⁴⁸, or vaccination with FimH⁴⁹, as such approaches would be ineffective at treating or preventing UTI caused by ST167. Together, these considerations form the basis of our criteria for the classification of ST167 as an atypical UPEC clone.

5. To avoid confusion for readers, it may be helpful to explicitly designate the capsule loci as K. pneumoniae loci throughout the text and figures and/or add some sort sub/super script annotation to the KL numbers.

We thank the reviewer for this pertinent observation and have changed the text with a superscript of Group 1. For example, KL124^{G1}; KL30^{G1}.

Response to comments from Reviewer 3.

i) while the authors have done a great job to change the language and discuss the caveats of the data, I remain nervous about the conclusions. The authors have noted that a large proportion of the data are derived from surveillance programs in the United States- which is good in the sense that these data will represent a more systematic sample, but in the context of the presented phylogeography and the breakdown of isolate sources per sub-clade, I wonder if the apparent association of ST167 with UTI is actually driven by capturing an ‘outbreak’ of clade C2 associated with UTI in the United States, rather than a global phenomenon that applies to the whole ST? (clade C2 is contributing the highest proportion of UTI isolates and is almost exclusively from the United States – with very short branch lengths in the tree. Other clades contribute a much lower proportion of UTI isolates). Have the authors explored the proportions of UTI vs other sources per ST for subsets of the data stratified by geography and/or subclade (sample sizes permitting)? Do the numbers support more widespread association of ST167 with UTI (even if the sample sizes are insufficient for statistical significance)? If not, perhaps the manuscript title, abstract and conclusions could be clarified to explicitly implicate that the C2 clade may be an emerging UPEC clone of concern, but is so far only detected in the United States?

We are encouraged by the positive comments from the reviewer concerning the changes made during revision and our discussion of the limitations of our data. The reviewer correctly points out that most sequenced isolates originate from the US, and we acknowledge this limitation in our revised manuscript. To address the reviewer's concern, we examined the proportion of ST167 isolates from each clade associated with UTI/urine vs blood and rectal sources in different geographical regions. The analysis confirms a widespread association of ST167 with UTI/urine, including strains from clade C2 (see Figure below). While this data was already included in Supplementary Data 2 (Genome Metadata for ST167 isolates), we believe the new figure will make it easier for readers to interpret this information. As noted by the reviewer, apart from the US, the sample sizes are insufficient for statistical analysis. This **new figure** is now included in the revised manuscript as Supplementary Figure 8.

Supplementary Fig 8. Column graphs describing the proportion of isolates from different sources for each phylogenetic clade and geographical region. The total number of isolates sourced from each geographical region (n) is noted in brackets. Isolate sources are colour coded according to the legend.

In addition to the analysis above, we also investigated the number of ST167 isolates sourced from urine in the SNP cluster PDS000063179.375 provided via the NCBI pathogen detection tool. This cluster encompasses the HC20 cluster 109409 of subclade C2. In this dataset, 233/528 isolates were sourced from urine, with a breakdown as follows: North America (n=212 urine isolates), France

(n=4), Lebanon (n=3), Germany (n=2), Mexico (n=1), China (n=1), Peru (n=1) and Japan (n=1). Although this dataset includes many genome sequenced ST167 isolates available in Enterobase, our analysis supports the widespread dissemination of subclade C2 across diverse global regions. To enhance clarity, we have **updated Supplementary Figure 9** to include the number of urine-derived ST167 isolates in this dataset (shown below).

Overall, the combined analysis shows that ST167 clade C2 strains isolated from UTI/urine are predominantly sequenced in the United States; however, similar strains have also been identified from UTI/urine sources in other countries. This indicates a broader geographic distribution beyond North America, and therefore we do not believe a major change in our conclusions is required.

Supplementary Fig 9. Bar graph depicting the number of isolates from NCBI pathogen detection SNP cluster PDS000063179.375 per country (total isolates = 528). Metadata for genomes can be found at NCBI pathogen detection (https://www.ncbi.nlm.nih.gov/pathogens/tree/#Escherichia_coli_Shigella/PDG000000004.5438/PDS000063179.466?accessions=PDT000899168.3). Isolates were clustered using whole genome MLST (wgMLST) with a clustering threshold of <25 allelic differences. The number of isolates sourced from urine is indicated.

ii) It is stated that the proportion of isolates that are from UTI for ST167 is not statistically different from the proportions in STs ST131, ST410, ST405, ST361, ST648 and ST1193. Are all of these STs considered major UPEC clones? Are any of the other STs (with lower proportion from UTI) considered UPEC clones? These answers will help to clarify if the trends seen in the data can be interpreted as the authors have implied.

ST131 and ST1193 (phylogroup B2), ST410 (phylogroup C), ST405 (phylogroup D), and ST648 (phylogroup F) are major antibiotic resistant UPEC clones associated with high rates of UTI and

bloodstream infection. Less is known about ST361 and we would not refer to it as a major UPEC clone.

In our manuscript, we note that the higher number of ST167 sequenced isolates is statistically significant (p value < 0.05) compared to ST69, ST38, ST10, ST162, ST58, ST297, ST88, ST349, ST504, ST48, ST3580, ST73 and ST95. Among these STs, ST69, ST73 and ST95 are major UPEC clones.

Based on the 3 reviewers' comments from our original submission, we have been careful in the way we interpret our data, as this represents numbers of isolates sequenced rather than epidemiological surveillance. We have therefore presented the proportion of ST167 isolates from UTI/urine sources compared to the proportion of isolates sequenced from other STs. Our interpretation is that:

L149-152: The increase in ST167 depositions to Enterobase, coupled with the high number of sequenced isolates from UTI/urine sources, suggests this ST represents an emergent phylogroup A uropathogenic clone.

To help address the reviewer's comments, the following changes were also made to the Supplementary figure 1 legend (changes underlined).

Supplementary Figure 1. Bar graph showing the 20 most deposited STs associated with human UTI/urine isolation in the Enterobase dataset in 2024. Each bar represents the number of isolates of an ST associated with an isolation source, red represents the number of isolates associated with human UTI/urine, and grey represents the number of isolates associated with all other sources. The percentage above each column represents the number of UTI/urine isolates divided by the total number of genomes in the respective ST. A pairwise Fisher's exact test for the proportion of UTI isolates from different sequence types did not lead to statistical significance (p value > 0.05) for ST167 compared to ST131, ST410, ST405, ST361, ST648 and ST1193 after correction of multiple comparisons by the Bonferroni correction. ST131 and ST1193 (phylogroup B2), ST410 (phylogroup C), ST405 (phylogroup D), and ST648 (phylogroup F) are major antibiotic resistant UPEC clones associated with high rates of UTI and bloodstream infection; less is known about ST361. In contrast, the increase in the number of ST167 sequenced isolates was statistically significant (p value < 0.05) compared to ST69, ST38, ST10, ST162, ST58, ST297, ST88, ST349, ST504, ST48, ST3580, ST73 and ST95 (Supplementary Table 1). Among these STs, ST69, ST73 and ST95 are major UPEC clones.

iii) I was very excited to see the inclusion of a meta-analysis to support the authors' findings, but as it stands it doesn't really add much to support the major claims in the paper. It seems that most of the studies in the meta-analysis are not explicitly focussed on UTI isolates- so it doesn't help to support the conclusions on uropathogenicity. Plus, the authors don't dig into the data in any depth. They state: "This analysis from 33 publications revealed that ST167 was the third most frequently detected sequence type across these studies (behind ST131 and ST410), underscoring the clinical relevance and expanding risk posed by this clone (Supplementary Data 7)" But looking at the underlying data in the supplementary table it's clear that not all studies were focussed on clinical isolates and there is substantial variation in the proportion of strains that are ST167 between studies (and geographies?). Since the sample sizes per study were widely varied, it doesn't really make much sense to use the aggregate rank as supporting evidence here. There may well be some useful evidence in the published data, but the current synthesis doesn't demonstrate this.

Our goal was to use the meta-analysis to compare the association of ST167 with UTI/urine against other STs. However, we acknowledge there are limitations in this approach that largely reflect our incomplete understanding of ST167 epidemiology, the primary issue addressed in this study. In our meta-analysis, 16/33 studies reported the identification of carbapenem-resistant ST167 isolates

from clinical samples. We also note that the precise origin of isolates is not uniformly reported across all studies. We agree that using aggregate rank is not appropriate for the interpretation of our meta-analysis. As such, we have toned down our interpretation to reflect the observation that ST167 was detected in comparable levels to ST410, ST405 and ST38, but at lower levels compared to ST131.

Modified text:

To further address potential database-driven biases, we conducted a meta-analysis of published studies from 2020 to 2025 that reported MLST data on carbapenem-resistant *E. coli* isolates, with a threshold of at least 50 isolates required for inclusion. The 33 studies included in this analysis varied widely in sample size and focus, with uneven geographical representation across regions, which limits the generalisability of the findings. Despite these constraints, the meta-analysis revealed that ST131 was the most frequently detected carbapenem-resistant ST, while ST167 was detected in comparable levels to ST410, ST405 and ST38 (Supplementary Data 7). The frequent detection of ST167 in clinical settings across multiple studies along with other high-risk clones supports our identification of carbapenem-resistant ST167 as an emerging threat.

iv) The mouse model data suggests that the ST167 strains are less fit in the UTI model than the comparator UPEC strain (?), and ST167 carry many fewer UTI-associated virulence factors than the comparator ST131 UPEC clone. The authors conclude that ST167 is therefore an atypical UPEC clone, but I don't see how these data lead to this conclusion without the context of the genomic analyses, and as described above I still have reservations about this. How would any other *E. coli* be expected to perform in the UTI model? And how common are these virulence factors in the broader *E. coli* population? (perhaps the authors can point to evidence in the literature here) Is there any evidence from the mouse model/ virulence factor data that suggest ST167 is more likely to cause UTI than any other *E. coli*?

The comparator UPEC strain used in the mouse UTI experiments is CFT073, a widely used UPEC reference strain from the ST73 clone. We have now noted that CFT073 belongs to ST73 in the revised manuscript.

Most UPEC strains belong to phylogroup B2 and D. Strains that do not belong to these phylogroups are rarely associated with UTI. ST167, the topic of this study, is an exception. ST167 belongs to phylogroup A and to the best of our knowledge no phylogroup A strains have been extensively studied for their capacity to colonise the mouse urinary tract.

The majority of the virulence factors examined in our study are largely specific to UPEC and are not common in other *E. coli* pathotypes. Our UPEC virulence factor list is a curated dataset that we assembled based on our knowledge of UPEC virulence. We provide a full list of the virulence factors in Supplementary Data 4, as well as a reference for each virulence factor. We believe this curated list will be a valuable resource for the field. To address the comment regarding a difference in virulence factors between UPEC strains from phylogroup B2 and D compared to ST167, we have added new data. Using our virulence factor list, we determined the median number of virulence genes in 100 randomly selected strains from the top sequenced STs in phylogroups B2 and D. Here, we show ST167 contains a reduced average number of virulence genes compared to strains from phylogroup B2 (ST73, ST12, ST127, ST95, ST998, ST372, ST131, ST141, ST428, ST1193) and phylogroup D (ST393, ST69, ST38, ST405, ST32, ST349, ST362, ST394), consistent with our detailed analysis of ST131 vs ST167.

Supplementary Figure 4. Bar graph depicting the median number of virulence genes for 100 randomly selected genomes from selected UPEC STs sourced from EnteroBase. The genomes belong to the 100ST dataset previously published⁴. STs are coloured by phylogroups as indicated. A virulence factor score was determined by summing the number of virulence genes identified. A list of the virulence genes is provided in Supplementary Data 4.

With regards to evidence from the mouse model/virulence factor data to suggest ST167 is more likely to cause UTI than any other *E. coli*. In fact, the opposite is true. We show almost all ST167 strains from clade B, C1 and C2 lack the genes encoding type 1 fimbriae. In all UPEC strains characterised to date, type 1 fimbriae play a critical role in UPEC bladder colonisation in mouse models. Thus, the inability to produce type 1 fimbriae suggests ST167 disease pathogenesis is different to previously characterised phylogroup B2 and D UPEC strains and associated with alternative mechanisms of colonisation and/or virulence factors. Indeed, our current knowledge of UPEC pathogenesis describes type 1 fimbriae as a canonical virulence factor, and thus based on this knowledge we cannot explain the high association of ST167 with UTI. This finding forms the basis of our criteria for the classification of ST167 as an atypical UPEC clone.

We also note that our description of ST167 as an atypical UPEC is analogous to seminal work that described atypical enteropathogenic *E. coli* (EPEC), which are distinguished from typical EPEC by the absence of bundle forming pili (PMID: 27571974).